# Contrasting radiation and soil heat fluxes in Arctic shrub and wet sedge tundra

Inge Juszak[1], Werner Eugster[2], Monique M. P. D. Heijmans[3], and Gabriela Schaepman-Strub[1]

[1]Department of Evolutionary Biology and Environmental Studies, University of Zurich, Winterthurerstrasse 190, 8057 Zurich, Switzerland

[2]Institute of Agricultural Sciences, ETH Zurich, Universitatstrasse 2, 8092 Zurich, Switzerland

[3]Plant Ecology and Nature Conservation, Wageningen University, Droevendaalsesteeg 3, 6708 PB Wageningen, The Netherlands

*Correspondence to:* I. Juszak (inge.juszak@gmx.de)

**Abstract.** Vegetation changes, such as shrub encroachment and wetland expansion, have been observed in many Arctic tundra regions. These changes feed back to permafrost and climate. Permafrost can be protected by soil shading through vegetation as it reduces the amount of solar energy available for thawing. Regional climate can be affected by a reduction in surface albedo as more energy is available for atmospheric and soil heating. Here, we compared the shortwave radiation budget of two common Arctic tundra vegetation types dominated by dwarf shrubs (*Betula nana*) and wet sedges (*Eriophorum angustifolium*) in North-East Siberia. We measured time series of the shortwave and longwave radiation budget above the canopy and transmitted radiation below the canopy. Additionally, we quantified soil temperature and heat flux as well as active layer thickness. The mean growing season albedo of dwarf shrubs was $0.15 \pm 0.01$, for sedges it was higher ($0.17 \pm 0.02$). Dwarf shrub transmittance was $0.36 \pm 0.07$ on average, and sedge transmittance was $0.28 \pm 0.08$. The standing dead leaves contributed strongly to the soil shading of wet sedges. Despite a lower albedo and less soil shading, the soil below dwarf shrubs conducted less heat resulting in a $17\,\mathrm{cm}$ shallower active layer as compared to sedges. This result was supported by additional, spatially distributed measurements of both vegetation types. Clouds were a major influencing factor for albedo and transmittance, particularly in sedge vegetation. Cloud cover reduced the albedo by 0.01 in dwarf shrubs and by 0.03 in sedges, while transmittance was increased by 0.08 and 0.10 in dwarf shrubs and sedges, respectively. Our results suggest that the observed deeper active layer below wet sedges is not primarily a result of the summer canopy radiation budget. Soil properties, such as soil albedo, moisture, and thermal conductivity, may be more influential, at least in our comparison between dwarf shrub vegetation on relatively dry patches and sedge vegetation with higher soil moisture.

## 1 Introduction

Recent climate warming in the Arctic (Stocker et al., 2013) is associated with increasing shrub abundance, cover, and biomass in many regions (Tape et al., 2006; Myers-Smith et al., 2011; Sturm et al., 2001b; McManus et al., 2012; Lantz et al., 2013; Frost and Epstein, 2014). However, vegetation can change in multiple directions and at larger scales the dominance of shrub tundra or wet sedge tundra is controlled by soil moisture and surface hydrology. While permafrost collapse leads to wetland

expansion in some continuous permafrost regions (Smith et al., 2005; Jorgenson et al., 2006; Lin et al., 2012; Schuur et al., 2015), drying has been observed in others (Oechel et al., 2000; Carroll et al., 2011; Jones et al., 2011; Lin et al., 2012). Shrub encroachment lowers the tundra albedo and thus positively feeds back to global warming (Sturm et al., 2001b; Lawrence and Swenson, 2011; Loranty and Goetz, 2012). However, larger scale atmospheric effects do not explain variations of permafrost

conditions at the local scale, where wetland vegetation is often associated with deeper active layers (Anisimov et al., 2002; Mi et al., 2014).

Dwarf birch (*Betula nana*) profits more than other species from warming (Walker et al., 2003a) and fertilisation (Bret-Harte et al., 2001; Hobbie et al., 2005). It is a common species in many Arctic regions (de Groot et al., 1997) and likely to be the primary driver of shrub expansion (Sturm et al., 2001a). Common cottongrass (*Eriophorum angustifolium*) is a widespread

wet sedge species (Phillips, 1954). In comparison with other sedges, *Eriophorum angustifolium* does not strongly profit from nutrient addition or warming (Shaver et al., 1998). However, it can expand in disturbed areas (Chapin and Shaver, 1981; Nauta et al., 2015) and where the surface gets wetter due to abrupt permafrost thaw (Schuur et al., 2015).

Arctic tundra ecosystems commonly comprise small scale vegetation patterns of shrubs, graminoids, and cryptogams associated with soil pH and moisture variation (Chapin et al., 2000b; Gamon et al., 2012). This intra-ecosystem variability is

relevant for the radiation budget as it can have stronger effects on the summer albedo than the difference among biomes, such as tundra and boreal forest (Chapin et al., 2000b; Eugster et al., 2000). Vegetation alters the radiation budget and turbulent energy fluxes at the soil surface which is critical for the ground heat flux and thus for permafrost thaw (Jacobsen and Hansen, 1999; Beringer et al., 2005). Shallower thaw depths have been observed below shrub canopies as compared to below other tundra vegetation (Anisimov et al., 2002; Walker et al., 2003b). Blok et al. (2010) suggested that the overall warming effect of

Arctic shrub encroachment due to decreasing albedo can be mitigated by soil shading.

The surface radiation budget is strongly influenced by cloud cover, which reduces the amount of incoming shortwave radiation and increases the fraction of diffuse light. High cloud fractions between 65% and 90% have been reported over Arctic land surfaces during the summer months (Curry et al., 1996; Wang and Key, 2005a; Dong et al., 2010). Recently, Arctic cloud cover has increased (Wang and Key, 2005b) and further increase is likely due to climate change (Chapin et al., 2005; Vavrus

et al., 2009). Cloudy conditions reduce the albedo at solar zenith angles of 60° or more (Yang et al., 2008) and increase the radiation fraction reaching the soil below the vegetation (at solar zenith angles above 50°, Mahat and Tarboton, 2012). Therefore, changes in cloud cover potentially impact the tundra surface radiation budget and interact with the predicted changes in tundra vegetation.

Furthermore, additional components of the plant–soil system are closely linked with the radiation budget. For example, it

has been shown that canopy shading affects the abundance and richness of mosses and lichens, which are suppressed by well-growing deciduous shrubs (van Wijk et al., 2003; Walker et al., 2006). Moreover, the radiation budget is linked with the carbon cycle. $CO_2$ fluxes were found to be highly related to net radiation in an Arctic tussock tundra site (Oechel et al., 2014).

Despite the importance of shading for the permafrost energy budget and plant species competition, it is rarely measured below different tundra vegetation types. While several studies assessed solar radiation transmittance below Arctic shrubs (Bewley

et al., 2007; Chong et al., 2012; Juszak et al., 2014; Williams et al., 2014), this study compared shrub shading with shading by

other vegetation types. Furthermore, the radiation and soil heat flux budget of Arctic tundra has been more extensively studied in Alaska, Canada, and Europe than in the vast Siberian lowlands.

In our study, we evaluate two main hypotheses. The first hypothesis is that the canopy radiation budget impacts permafrost thaw. The second hypothesis is that weather conditions significantly affect radiation fluxes in Arctic tundra. We compare two widespread and discriminative tundra vegetation types to assess these hypotheses, dwarf shrubs and wet sedges. For these types, we quantified the above-canopy radiation budget, below-canopy transmitted shortwave radiation, and soil heat fluxes. We complemented time series measurements of these three components with spatially distributed measurements at the Kytalyk research site, North-East Siberia. Our results will assist modelling attempts in providing details on local scale variability of albedo, soil shading, and soil heat flux in an Arctic tundra ecosystem.

## 2 Methods

### 2.1 Field site, vegetation, and soil

The study area is located in a drained thaw lake bed in the Kytalyk nature reserve, North-East Siberia (70.83 °N, 147.49 °E, Fig. 1a). It is characterised by continuous permafrost and a shallow soil layer which thaws every summer and refreezes in winter. The depth of this layer, called active layer thickness, varies in the range of 15–55 cm (van Huissteden et al., 2005) at the study site. The mean annual permafrost temperature at 15 m depth close to the site is $-9.4\,°C$ (Romanovsky et al., 2010). The study region in lowland tundra is underlain by very ice-rich permafrost (Iwahana et al., 2014), which makes it susceptible to rapid changes in case of warming (Jorgenson et al., 2006). A multi-year study by Parmentier et al. (2011a) observed the snowmelt between 18 May and 10 June. The growing season started about four weeks after snowmelt and ended early September in all years (Parmentier et al., 2011a). Ecosystem-scale measurements of energy and carbon fluxes are available from flux tower measurements by the VU Amsterdam (van der Molen et al., 2007; Parmentier et al., 2011b; Budishchev et al., 2014, site Russia, Chokurdakh, http://www.europe-fluxdata.eu).

The vegetation at the study site is classified as tussock sedge, dwarf shrub, and moss tundra in the Circumpolar Arctic vegetation map (Walker et al., 2005). More specifically, in the drained thaw lake bed, elongated, well-drained patches of erect dwarf shrub tundra alternate with depressions of sedge, moss, dwarf shrub wetland (Fig. 1b). Dwarf shrub and wet sedge patches are irregularly shaped and about 10–20 m wide and 70–150 m long (Fig. 1b). The surface elevation of dwarf shrub patches is 0.3–0.7 m higher than of wet sedge depressions. In lowland tundra, the continuously high water content at the wet sedges can be replenished by lateral water fluxes due to microtopography and by thawing the ice-rich active layer (Helbig et al., 2013).

The centre of dwarf shrub patches is dominated by dwarf birch (*Betula nana*, Fig. 1e, f). Willows (mainly *Salix pulchra*) complement the dwarf birch and dominate the canopies along the rivers. Communities of lingonberry (*Vaccinium vitis-idaea*), mosses, and lichen surround the dense dwarf birch vegetation. Towards the lower, wetter areas peat mosses (Sphagnum spp.) and sedges border the dwarf shrub patches. Most commonly, the wet sedge community is comprised of common cottongrass (*Eriophorum angustifolium*, Fig. 1c, d), which does not form tussocks. Although dwarf birch dominated areas are usually

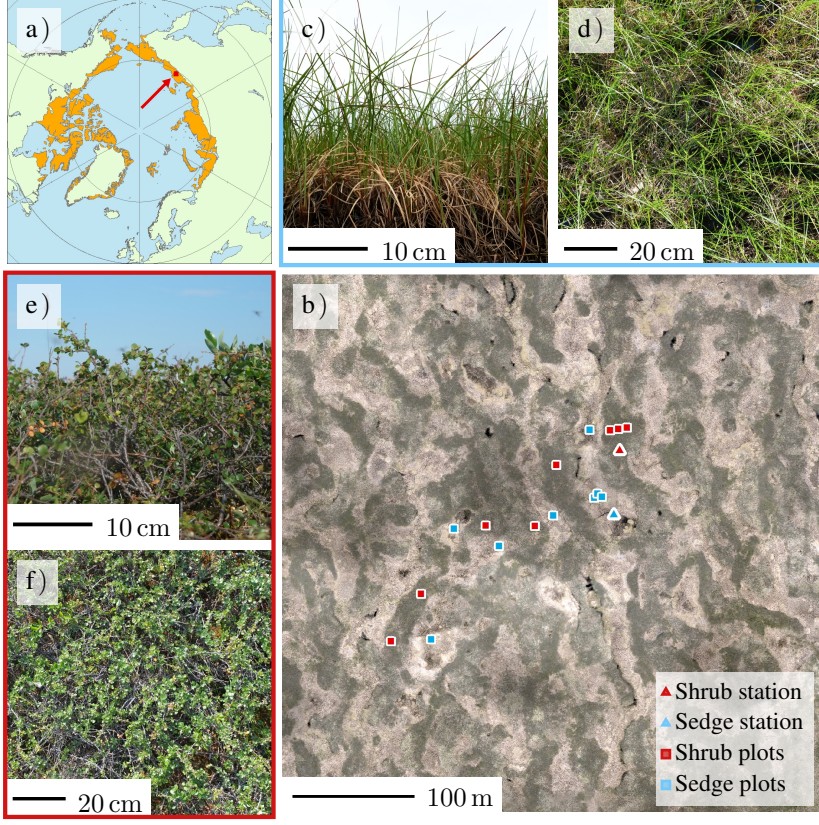

**Figure 1.** (a) Location of the Kytalyk research station and Arctic tundra extent (data from Walker et al., 2005), (b) red-green-blue orthomosaic of the site (drone flight July 2014) where shrubs appear as dark surfaces, while sedges are represented by bright shades including the locations of stations for time series and distributed plot measurements on dwarf shrubs and wet sedges, (c, d) sedge vegetation and (e, f) dwarf shrub vegetation.

separated from wet sedge areas by the described transition zones, in some places they can be found directly bordering each other. Sedges can invade drowning shrub-covered areas (Myers-Smith et al., 2011), especially after disturbance (Nauta et al., 2015), and shrubs can colonise peat moss covered areas, which in turn invade wet sedge depressions. Details on the canopy and soil characteristics of dwarf shrub and wet sedge vegetation are provided in Table 1.

## 2.2 Measurements

We assessed the effect of wet sedge versus dwarf shrub dominated vegetation on radiation fluxes above and below canopy as well as the soil heat flux at $10\,\mathrm{cm}$ depth with field measurements. Time series of radiation and soil heat flux were acquired at a permanent location close to the centre of one patch per vegetation type. These measurements were complemented by sporadic measurements in eight spatially distributed plots per vegetation type to assess the spatial variation. For our below

**Table 1.** Vegetation and soil characteristics of dwarf shrubs and wet sedges; all data were collected in our study, except the depth of soil layers; the measurements are described below in the methods section; the dwarf shrub area index is the projected area index adjusted by the average measurement efficiency.

| Characteristic | Dwarf shrubs | Wet sedges |
| --- | --- | --- |
| Dominant species | *Betula nana* | *Eriophorum angustifolium* |
| End of summer active layer thickness | 16–25 cm | 31–41 cm |
| End of winter snow depth | 32 cm | 71 cm |
| Canopy height | 17–32 cm | 38–66 cm |
| Standing dead leaf height | - | 12–34 cm |
| Area index | 0.7–1.2 green leaves, 0.4–1.1 wood | 1.0–2.1 green leaves, 1.1–2.3 standing dead leaves |
| Dry biomass | $86 \, \mathrm{g\,m^{-2}}$ green leaves, $1183 \, \mathrm{g\,m^{-2}}$ wood | $181 \, \mathrm{g\,m^{-2}}$ green leaves, $162 \, \mathrm{g\,m^{-2}}$ standing dead leaves |
| Background | 56% shrub litter, 38% mosses (predominantly *Dicranum* sp., *Polytrichum* sp., and *Aulacomnium* sp.), 4% lichen | Water, wet litter |
| Depth of soil layers | 4–5 cm mosses, 10–15 cm highly organic soil, mineral clay soil mixed with organic matter[*1] | 13–19 cm water saturated, loose organic material, mineral clay soil mixed with organic matter[*2] |
| Volumetric soil moisture | $0.3$–$0.6 \, \mathrm{m^3\,m^{-3}}$ | $0.7 \, \mathrm{m^3\,m^{-3}}$ (saturated) |
| Thermal conductivity | $0.08 \, \mathrm{W\,m^{-1}\,K^{-1}}$ top soil, $0.80 \, \mathrm{W\,m^{-1}\,K^{-1}}$ mineral soil | $0.44 \, \mathrm{W\,m^{-1}\,K^{-1}}$ top soil |
| Volumetric heat capacity | $0.5 \, \mathrm{MJ\,m^{-3}\,K^{-1}}$ top soil, $2.1 \, \mathrm{MJ\,m^{-3}\,K^{-1}}$ mineral soil | $3.3 \, \mathrm{MJ\,m^{-3}\,K^{-1}}$ top soil |

[*1] Blok et al. (2010)

[*2] Bartholomeus et al. (2012)

ground measurements in the wet sedge plot, we defined the top of the dark, wet, and cohesive litter as reference height. In the dwarf shrub plot, we used the top of the moss or litter as reference height (Blok et al., 2011).

The time series were recorded by two automatic measurement stations on a dwarf shrub and an adjacent wet sedge patch, located about $50 \, \mathrm{m}$ apart (Fig. 1b, triangles). The instrument height was about $1.5 \, \mathrm{m}$ above canopy to ensure that the instrument's footprint covered only one vegetation type. We used Kipp & Zonen CMP11 pyranometers (285–2800 nm) for incoming and reflected shortwave radiation, and an array of four (on sedge) and five (on dwarf shrub) Kipp & Zonen SPLITE2 silicon pyranometers (400–1100 nm) for below-canopy transmitted shortwave radiation. We installed the instruments on the moss or litter surface below dwarf shrubs and below some of the sedge standing dead leaves but above the early summer water level. We measured net longwave radiation with a Kipp & Zonen CNR2 net radiometer (300–2800 nm and 4.5–42 µm) in each plot.

The two shortwave radiation flux components of the CNR2 also allowed for cross-validation with the CMP11 sensor data in our quality control procedure. Additionally, we cross-validated our incoming shortwave radiation measurements with one SPLITE2 pyranometer. Soil heat flux was measured in the organic top soil using three heat flux plates (HFP01, Hukseflux) per vegetation type at a depth of $10\,\text{cm}$ below the reference height. Soil temperature data were acquired using three sensors (T107, Campbell Scientific) per vegetation type at a depth of $4\,\text{cm}$ below the reference height. We measured soil moisture with two sensors (ThetaProbe ML2x, Delta-T Devices) per vegetation type and converted the signal to volumetric water content using standard parameters for organic soil. The data from all sensors were recorded every $30\,\text{sec}$ and averaged at $10\,\text{min}$ intervals using a Campbell Scientific CR1000 datalogger. The radiation data series covers 07 July 2013 – 31 August 2013 and 11 May 2014 – 17 August 2014. The soil flux data begin ten days later in 2013 and span the same period as radiation measurements in 2014. We measured the soil thermal properties with a KD2 PRO, Decagon Devices, instrument on 04 and 05 August 2013. The measurements were done at all locations of soil heat flux measurements and in one soil pit per vegetation type to estimate soil properties below the highly organic horizon. The measurement date was after a dry summer period.

In order to assess the spatial variability of dwarf shrub and wet sedge vegetation and the spatial representativeness of the time series measurements, we additionally measured vegetation and radiation parameters in eight plots of $1\,\text{m}^2$ per vegetation type (Fig. 1b, squares). Three of the eight plots were located within the same vegetation patch as the time series measurements but outside of the footprint of the instruments. In the 16 plots, we measured spectral exitance about $1\,\text{m}$ above the canopy with an Ocean Optics Jaz spectrometer using a bare $100\,\mu\text{m}$ fiber. Spectral irradiance was measured before and after the exitance measurements of each plot using the same spectrometer and an upwards pointing fiber equipped with a cosine corrector. From these two measurements we calculated the hemispherical-conical reflectance factor in nadir direction in the range of photosynthetically active radiation (PAR, 400–700 nm). We measured incident and transmitted PAR with a Delta-T Devices SunScan ceptometer. Below canopy, 64 sensors on a $1\,\text{m}$ long probe recorded transmitted PAR simultaneously with the above-canopy BF3 sensor. Active layer thickness and canopy height were measured relative to the reference height by inserting a metal probe at 25 points of a regular grid in every plot. Canopy height was estimated as average height of the highest leaves within a $5\,\text{cm}$ radius around the measurement point. Additionally, we measured maximum height at all plots. We measured projected dwarf shrub leaf and wood area using a $1\,\text{m}^2$ point quadrat and recording all contacts between a vertically inserted needle and the vegetation at 81 points (Wilson, 1959). The leaf area of other vascular plants on the dwarf shrub plots was negligible. We used two additional plots to measure dwarf shrub biomass after cutting all shrubs, separating leaves from wood, and air-drying the biomass for about one week. We scanned a subset of the leaves and wood and estimated the leaf area index (LAI). We found that the point quadrat counts underestimated the shrub leaf area by 15% and 28%, which was 0.18 and 0.25 in absolute values. Dwarf shrub wood area was overestimated by 5% and underestimated by 17% using point quadrats in comparison with the destructive measurements (0.04 and 0.14 difference in absolute values).

To estimate sedge green leaf area index non destructively, we used an empirical allometric approach,

$$\text{LAI} = \frac{h_c}{h_r} \cdot \frac{1}{A} \cdot \sum_{i=1}^{m} n_i \cdot L_i$$

where $h_c$ is the canopy height, $A$ is the size of the investigation area (m$^2$), $n$ is the number of tillers of size $i$, and $L$ is the average green leaf area (m$^2$) of tillers of size $i$. Tiller size is expressed by the number of leaves that a tiller has, so the smallest tiller with $i = 1$ has one leaf only and the largest tillers have $m = 7$ leaves. The allometric value $L_i$ for each tiller size was determined via destructive sampling on a $50 \cdot 50 \, \text{cm}^2$ plot with reference canopy height $h_r$ from which 18 tillers out of 162 were randomly selected for analysis. We measured the length and width of all leaves and subsequently cut the leaves in segments to allow for accurate scanning of the one-sided leaf area. Thus, the empirical relationship of LAI as a function of $n_i$ and $h_c$ could be used on eight $1 \, \text{m}^2$ plots for non destructive LAI estimation. Within each plot, we selected 16 subplots of $10 \cdot 10 \, \text{cm}^2$ size. In each of them we counted the number of tillers of each tiller size class ($n_i$) and measured the canopy height $h_c$ to estimate LAI. For validation, three destructive harvests were done to ascertain the quality of the non destructive LAI estimates based on weighing dry biomass of green and standing dead leaves separately and scanning a subset. This indicated that green LAI was accurate to within $\pm 0.4 \, \text{m}^2 \, \text{m}^{-2}$: tiller counting overestimated the LAI by 0.4 and 0.3 in two plots while LAI was underestimated by 0.4 on the third plot.

## 2.3 Data analysis

To quantify the effects of vegetation type on the radiation budget, we computed net radiation $R_n$,

$$R_n = K_\downarrow - K_\uparrow + L_\downarrow - L_\uparrow$$

with $K$ and $L$ being shortwave and longwave radiation fluxes, respectively. Arrows in the index show downward ($\downarrow$) and upward ($\uparrow$) directed radiation. The difference $K_\downarrow - K_\uparrow$ is the net shortwave radiation. Shortwave albedo $\alpha$ is derived as

$$\alpha = \frac{K_\uparrow}{K_\downarrow}$$

and transmittance $\tau$ is the ratio between downwelling shortwave radiation measured below canopy (index $bc$) and the same measurement carried out above canopy (index $ac$),

$$\tau = \frac{K_{\downarrow,bc}}{K_{\downarrow,ac}}.$$

To assess transmittance of each vegetation type, we combined all data of the five (below dwarf shrubs) and four (below sedges) sensors.

The radiation budget is strongly influenced by weather conditions and the solar zenith angle. We estimated both in order to isolate the vegetation effects. We calculated the solar zenith angle of each measurement using a MATLAB® script by Vincent Roy following an algorithm by Reda and Andreas (2004). We binned all solar zenith angles into $2°$ bins and used $10 \, \text{min}$ averages for $K$ and $L$ to compute $R_n$, $\alpha$, and $\tau$. Additionally, the average effect of vegetation type and cloud cover on the radiation budget was estimated independently of the solar angle. In this case, we reduced the solar angle influence by taking daily average fluxes of $K$ and $L$ to compute the mean and standard deviation of $R_n$, $\alpha$, and $\tau$. We used the term 'soil shading' as reduction of incoming shortwave radiation (1–transmittance). We defined the peak growing season as 1 July – 15 August (Fig. 2).

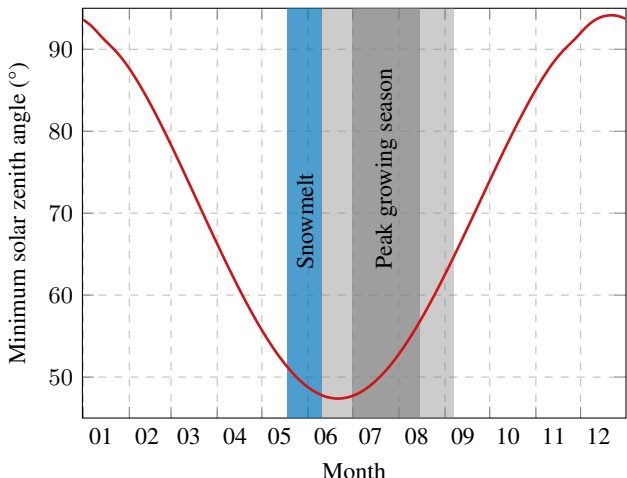

**Figure 2.** Variation of the solar zenith angle at solar noon (14:10 local time); snowmelt (blue) and growing season (light grey) in different years (dates from Parmentier et al., 2011a) and peak growing season (1 July – 15 August).

In order to quantify the effects of clouds on the shortwave radiation budget, we classified the cloud cover into three categories, clear sky, partly cloudy, and cloudy. The classification was based on the cloud factor $cf$ (Tuller, 1976; Crawford and Duchon, 1999)

$$cf = 1 - \frac{K_{\downarrow,measured}}{K_{\downarrow,potential}}$$

with measured ($K_{\downarrow,measured}$) and potential ($K_{\downarrow,potential}$) incoming shortwave radiation. We computed $K_{\downarrow,potential}$ for each 10 min interval using an atmospheric transfer model by Corripio (2003) on the basis of the Iqbal (1983) study on solar radiation transfer through the atmosphere. The most important model inputs were site location, measured air temperature, and measured surface albedo. Other parameters were ozone layer thickness (300 DU), visibility (180 km), and relative humidity (80%). Topographic shading was neglected because the research site is almost flat. We validated the model using observed clear-sky

$K_{\downarrow,measured}$. As the relative error of $K_{\downarrow,measured}$ and $K_{\downarrow,potential}$ increases at low values in the morning and evening, we only computed cloud factors when $K_{\downarrow,potential} > 50\,\mathrm{W\,m^{-2}}$. We calculated the mean cloud factor, either within a day or of each time step. While we used the daily classification to analyse vegetation and cloud impacts, the 10 min classification was needed for solar zenith angle effects. Each day with a mean cloud factor below 0.15 was classified as 'Clear sky', days above 0.55 were classified as 'Cloudy' while all other days were categorised as 'Partly cloudy'. We used the same thresholds as for

daily values for the 10 min intervals but with the additional condition that clear-sky and cloudy intervals required a standard deviation $< 0.1$ determined from 1-hour centred at the 10 min interval of interest. Higher variation indicated partly cloudy conditions. We used MATLAB® for all computations.

We used t-tests to assess the difference between dwarf shrub and sedge characteristics, namely in canopy height, LAI, PAR reflectance, and PAR transmittance on the spatially distributed plots. Mean values are shown $\pm$ standard deviation to illustrate

the spatial or temporal variability.

## 3 Results

### 3.1 Canopy structure

Dwarf shrub and sedge vegetation showed different canopy characteristics, radiation budgets, and soil heat fluxes. The sedge canopy was on average $48 \pm 8\,cm$ high, almost twice the height of the dwarf shrub canopy (Fig. 3a). The estimated sedge green leaf area index was on average $1.4 \pm 0.3$ and the projected dwarf shrub leaf area index as estimated using point quadrats was $0.8 \pm 0.1$ (Fig. 3b). Apart from the green leaves, also wood and standing dead material can influence the radiation budget. The dwarf shrub wood area index was similar to the leaf area index. The sum of projected shrub leaf and wood area index was $1.5 \pm 0.3$, slightly higher than wet sedges green leaf area index. In all three destructive harvests of wet sedge leaf and standing dead leaf area, we found that standing dead leaf area was 1.1 times green leaf area.

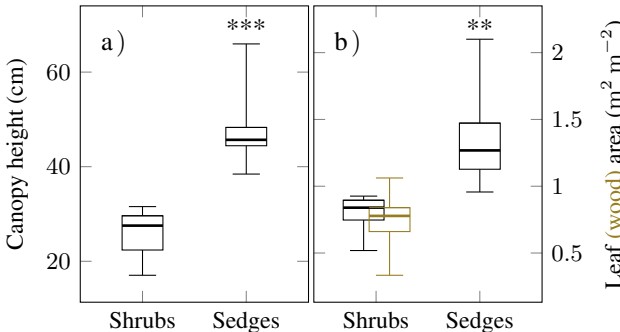

**Figure 3.** (a) Spatial variability of canopy height and (b) green leaf and wood area index measured 28 – 30 July 2013, percentiles (25, 50, and 75), minimum and maximum values of eight plots per vegetation type; please note that the wet sedges canopy additionally comprises standing dead leaves with an area of about 1.1 times green leaf area; significant differences between vegetation types are shown with ** (p $\leq$ 0.01) and *** (p $\leq$ 0.001).

### 3.2 Above-canopy radiation budget

Dwarf shrub and wet sedge vegetation influence the radiation budget differently. While the dwarf shrub canopy reflected less shortwave radiation, it emitted more longwave radiation (Fig. 4a, b). The difference between both vegetation types in net radiation on average and at any solar zenith angle was very small (Table 2, Fig. 4c, 6a, b).

During the growing season, sedge albedo was consistently higher than dwarf shrub albedo (Fig. 5, 6c, d). The growing season mean daily albedo was 0.15 for dwarf shrubs and 0.17 for sedges (Table 2). In absolute terms, the dwarf shrub vegetation–soil system absorbed on average $5\,W\,m^{-2}$ more shortwave radiation than sedge vegetation during the 2013 and 2014 growing seasons. The growing season albedo differences between the vegetation types based on time series measurements are consistent with spatially replicated spectrometer measurements in the PAR region in eight plots per vegetation type (Fig. 7b). However,

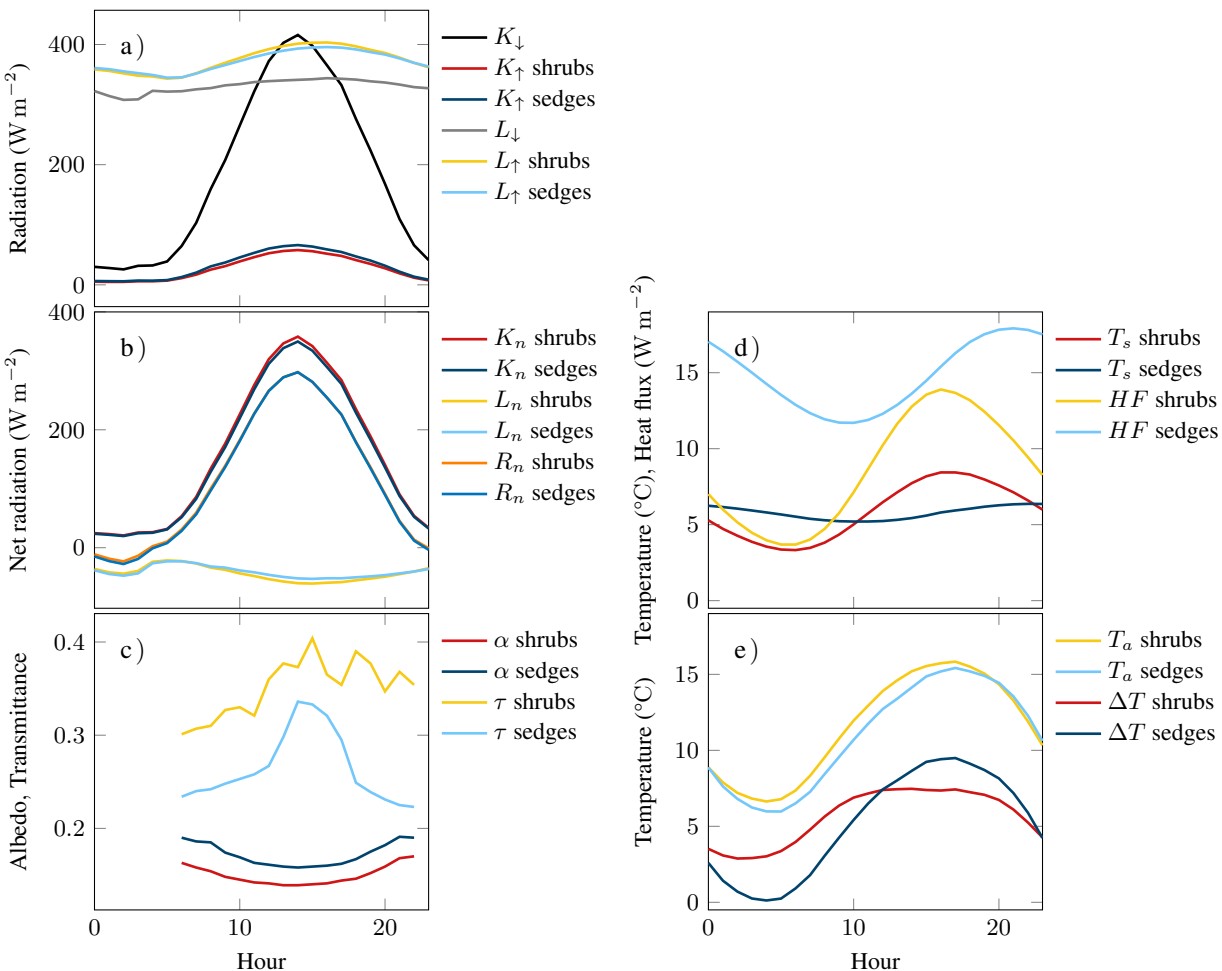

**Figure 4.** (a) Average diel cycle of above canopy shortwave ($K$) and longwave ($L$) radiation fluxes, (b) above canopy net shortwave ($K_n$), net longwave ($L_n$) and net ($R_n$) radiation, (c) albedo ($\alpha$) and transmittance ($\tau$), (d) soil temperature ($T_s$) at $4\,\mathrm{cm}$ depth and soil heat flux ($HF$) at $10\,\mathrm{cm}$ depth, and (e) air temperature at $1.7\,\mathrm{m}$ above the soil surface ($T_a$) and gradient between air and soil temperature ($\Delta T$) of dwarf shrubs and wet sedges during the growing season; solar noon at 14:00 local time.

PAR reflectance was $0.024 \pm 0.006$ above dwarf shrubs and $0.034 \pm 0.008$ above sedges and thus much lower than shortwave albedo.

Additionally, the spring snow depth was $39\,\mathrm{cm}$ deeper in the sedge depression than on the elevated dwarf shrub patch (Table 1). Thus, the snow on the sedge patch melted about ten days later (03 June 2014) as compared to the dwarf shrub patch (24 May 2014). Due to the albedo difference between snow covered and snow free surfaces, the dwarf shrub patch absorbed $125\,\mathrm{MJ\,m^{-2}}$ ($145\,\mathrm{W\,m^{-2}}$ on 10 days) shortwave radiation more than the sedge patch in this time (Fig. 7a).

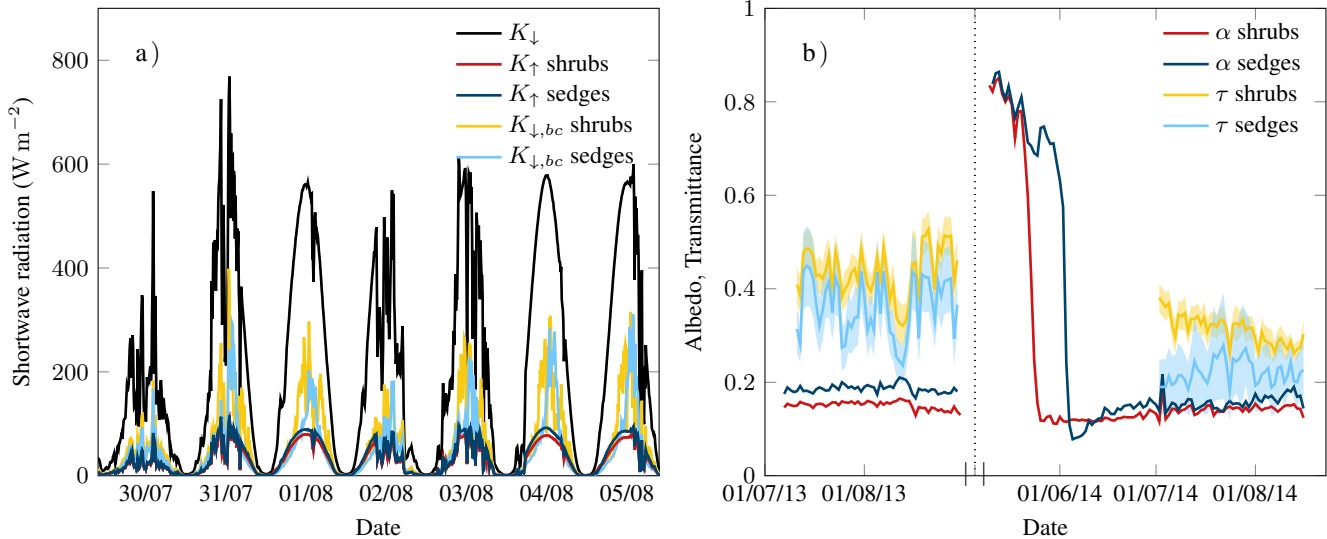

**Figure 5.** (a) Shortwave radiation fluxes incoming ($K_\downarrow$), reflected ($K_\uparrow$) and below canopy ($K_{\downarrow,bc}$), one week time series of 2014 and (b) daily albedo ($\alpha$) and transmittance ($\tau$) time series; shaded area around transmittance represents $\pm$ standard error of the mean of the spatial replicates.

**Table 2.** Energy fluxes and soil temperatures (mean $\pm$ standard deviation determined from daily averages) of dwarf shrub and wet sedge vegetation under varying cloud conditions in the peak growing season 2013 and 2014; sw. denotes shortwave.

| Type | Condition | Albedo | Transmittance | Net radiation $\mathrm{W\,m^{-2}}$ | Net sw. radiation $\mathrm{W\,m^{-2}}$ | Soil temp. ($-4\,\mathrm{cm}$) °C | Soil heat flux ($-10\,\mathrm{cm}$) $\mathrm{W\,m^{-2}}$ |
|---|---|---|---|---|---|---|---|
| Dwarf shrubs | All | $0.15 \pm 0.01$ | $0.36 \pm 0.07$ | $116 \pm 38$ | $157 \pm 54$ | $5.9 \pm 2.0$ | $8.6 \pm 3.2$ |
| | Clear sky | $0.15 \pm 0.01$ | $0.32 \pm 0.04$ | $144 \pm 34$ | $215 \pm 34$ | $7.2 \pm 2.0$ | $10.6 \pm 3.7$ |
| | Partly cloudy | $0.15 \pm 0.01$ | $0.36 \pm 0.06$ | $123 \pm 30$ | $164 \pm 36$ | $6.1 \pm 1.7$ | $8.8 \pm 2.9$ |
| | Cloudy | $0.14 \pm 0.02$ | $0.40 \pm 0.07$ | $75 \pm 27$ | $89 \pm 26$ | $4.3 \pm 1.6$ | $6.1 \pm 2.0$ |
| Wet sedges | All | $0.17 \pm 0.02$ | $0.28 \pm 0.08$ | $114 \pm 38$ | $152 \pm 52$ | $5.8 \pm 2.3$ | $14.8 \pm 5.2$ |
| | Clear sky | $0.19 \pm 0.02$ | $0.23 \pm 0.03$ | $140 \pm 34$ | $207 \pm 34$ | $7.0 \pm 2.6$ | $16.9 \pm 6.0$ |
| | Partly cloudy | $0.17 \pm 0.02$ | $0.27 \pm 0.07$ | $121 \pm 30$ | $160 \pm 36$ | $5.8 \pm 2.1$ | $15.1 \pm 5.0$ |
| | Cloudy | $0.16 \pm 0.02$ | $0.33 \pm 0.10$ | $74 \pm 27$ | $87 \pm 26$ | $4.6 \pm 1.8$ | $11.9 \pm 3.9$ |

Clouds reduced albedo and net radiation of both canopies throughout the summer (Fig. 6). The reduction in albedo was more pronounced for sedges than for dwarf shrubs (Table 2). While the clear-sky albedo of both vegetation types increased at higher solar zenith angles, the cloudy-sky albedo was similar to the clear-sky albedo at solar noon for all zenith angles (Fig. 6c, d). Due to the strong effect of zenith angle on clear-sky albedo, the cloud effects were strongest in August, when the minimum solar

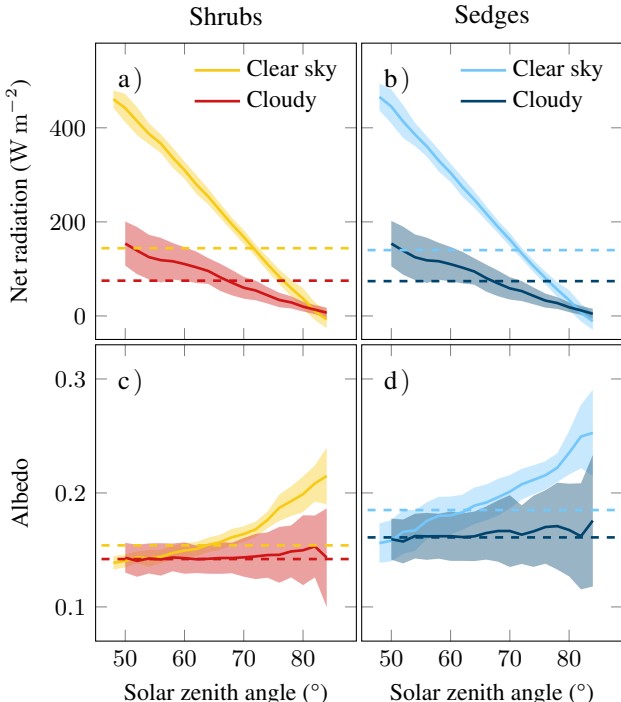

**Figure 6.** Dependence of (a, b) net radiation and (c, d) albedo on solar zenith angle and cloud cover for (a, c) dwarf shrub and (b, d) wet sedge, growing season mean $\pm$ standard deviation values calculated for 2° intervals; the dashed lines represent the mean diel value under each condition.

zenith angle was larger than in June or July (Fig. 2). In both summers 2013 and 2014 about 20% of the days were classified as clear sky and 20% as cloudy, the remaining as partly cloudy.

### 3.3 Soil shading

Canopy transmittance was on average 0.36 below dwarf shrubs and 0.28 below wet sedges during the growing season (Table 2, Fig. 4c, 8a, b). This difference implied that the surface below dwarf shrubs was exposed to $15\,\mathrm{W\,m^{-2}}$ shortwave radiation in addition to what we observed below sedge vegetation. The spatially distributed measurements in the PAR range also showed a significant difference between the two vegetation types (Fig. 8c). However, the major effect could be attributed to the multi-year standing dead leaves below the green sedge leaves. The green leaves transmitted more light $(0.62 \pm 0.11)$ than dwarf shrubs $(0.25 \pm 0.07)$, but most light was reflected or absorbed by the standing dead layer (Fig. 8c).

Transmittance was strongly influenced by clouds (Fig. 8a, b). On average, clouds increased the transmittance by 25% below dwarf shrubs and by 43% below sedges (Table 2). However, for specific locations and sun angles the clear-sky transmittance exceeded the transmittance of cloudy times. Transmittance was generally higher at small solar zenith angles, especially for

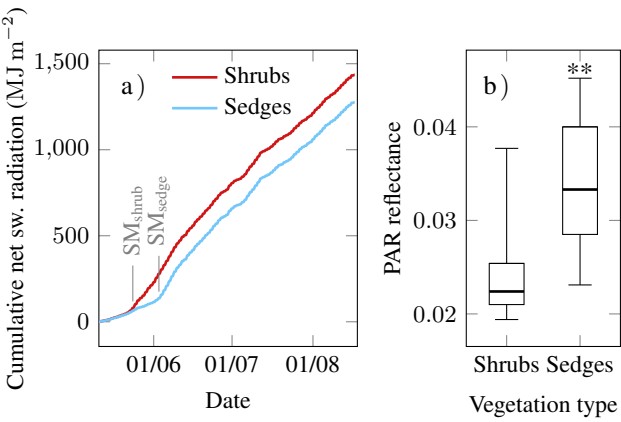

**Figure 7.** (a) Cumulative net shortwave (sw.) radiation 11 May – 16 August 2014, day of complete snowmelt (SM) indicated for dwarf shrubs and wet sedges and (b) spatial variability of PAR (400–700 nm) reflectance within vegetation type measured on 25 July 2013, percentiles (25, 50, and 75), minimum and maximum values of eight plots per vegetation type; significant differences between vegetation types are shown with ** (p ≤ 0.01).

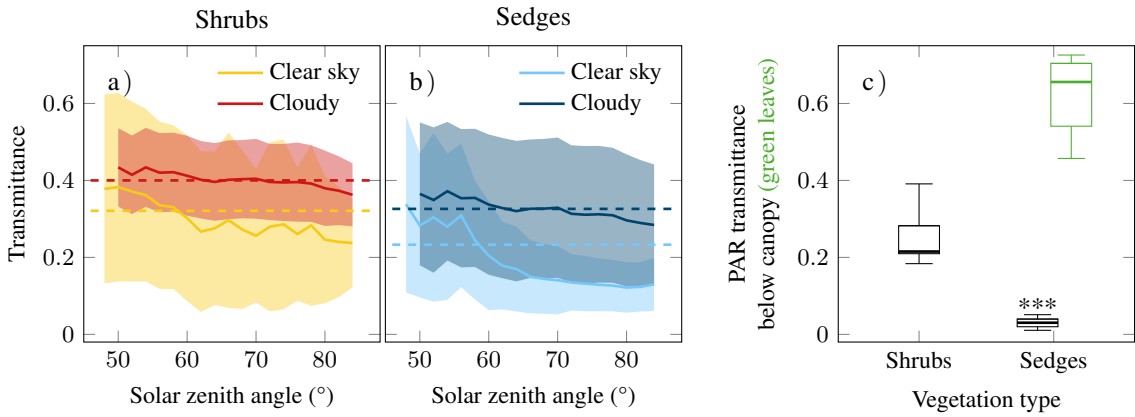

**Figure 8.** (a,b) Shortwave transmittance depending on solar zenith angle and cloud cover for dwarf shrub (a) and wet sedge (b), growing season mean ± standard deviation calculated for 2° intervals, the dashed lines represent the mean diel value under each condition and (c) spatial variability of PAR (400–700 nm) transmittance measured 03 August 2013, percentiles (25, 50, and 75), minimum and maximum values of eight plots per vegetation type; significant differences between vegetation types are shown with *** (p ≤ 0.001).

sedges and at clear-sky conditions (Fig. 8a, b). Furthermore, soil shading was highly spatially and temporally variable, especially under clear-sky conditions (Fig. 8a, b).

### 3.4  Soil heat flux and permafrost active layer

Except during the cold and snow-covered period, the soil heat flux at $10\,\mathrm{cm}$ depth was consistently higher below wet sedges than below dwarf shrubs (Fig. 4d, 9a). As soon as the dwarf shrub patch was partly snow free in May 2014, the soil heat flux at the sedges reached a peak of $30\,\mathrm{W\,m^{-2}}$ while the heat flux below dwarf shrubs was less than $5\,\mathrm{W\,m^{-2}}$. Afterwards, the
5  sedge soil heat flux reduced to about 1.6 times the flux below dwarf shrubs by the end of July (Fig. 9a). The mean growing season soil heat flux of 2013 and 2014 was $8.6\,\mathrm{W\,m^{-2}}$ on the dwarf shrub and $14.8\,\mathrm{W\,m^{-2}}$ on the sedge patch (Table 2). The 2014 growing season was wetter than 2013, resulting in elevated soil moisture content below dwarf shrubs ($0.52\,\mathrm{m^3\,m^{-3}}$ and $0.37\,\mathrm{m^3\,m^{-3}}$ in 2014 and 2013, respectively). The sedges soil was saturated at all times with a moisture content of about $0.7\,\mathrm{m^3\,m^{-3}}$ (Table 1). However, we observed higher water levels at the sedges in 2014. The wetter conditions in 2014 fostered
10  higher soil heat fluxes below both vegetation types. Top soil temperature below dwarf shrubs was on average $1.1\,^{\circ}\mathrm{C}$ warmer in the dry growing season 2013, while it was $0.6\,^{\circ}\mathrm{C}$ colder than below sedges in the wet growing season 2014. On average over both summer measurement periods, the top soil temperature was almost equal below both vegetation types (Table 2). The spatially distributed active layer thickness measurements were consistent with the soil heat flux measurements. On average, the active layer below sedges was 1.8 times deeper than below dwarf shrubs (Fig. 9b).

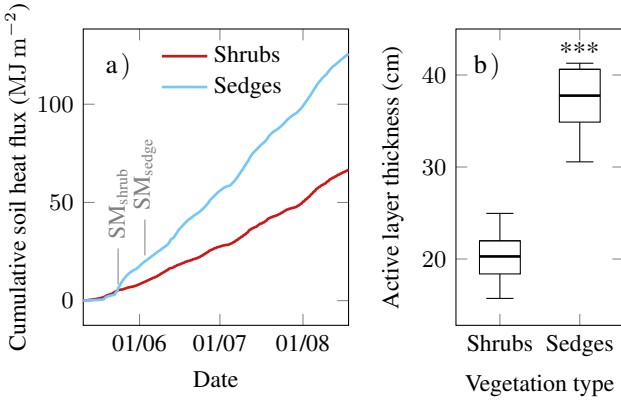

**Figure 9.** (a) Cumulative soil heat flux 2014 at $-10\,\mathrm{cm}$ depth, day of complete snowmelt (SM) indicated for dwarf shrubs and wet sedges and (b) spatial variability of active layer thickness within vegetation type, measured $28-30$ July 2013, percentiles ($25, 50,$ and $75$), minimum and maximum values of eight plots per vegetation type; significant differences between vegetation types are shown with *** ($p \le 0.001$).

## 4  Discussion

We found that wet sedges shade the soil more efficiently than dwarf shrubs, which is in contrast to the higher soil heat flux below sedges. The considerable shading by wet sedges can partly be explained by the thick layer of standing dead leaves. The soil heat flux, on the other hand, depended strongly on soil thermal properties. This is in contrast to the studies by Blok et al. (2010) and Briggs et al. (2014), which identified soil shading as important control of local permafrost thaw. A schematic of the

differences we found between key components of the dwarf shrub and wet sedge system can be found in Fig. 10. The different components are explained in more detail below.

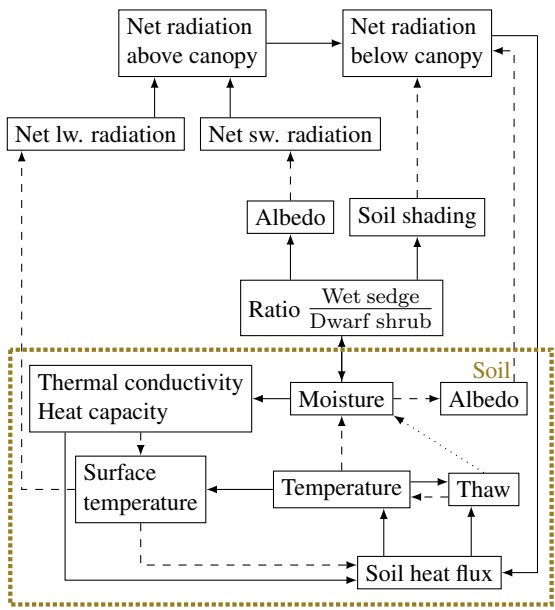

**Figure 10.** Different vegetation types are associated with soil properties and soil heat fluxes (brown box), the radiation budget above and below canopy. Latent and sensible heat fluxes are not included as they were not measured in this study and effect directions are unclear. Permafrost thaw can feed back to soil moisture, but the direction of the effect depends on ice content and drainage conditions. Solid arrows represent positive, dashed arrows negative, and dotted arrows unknown feedback; lw. and sw. denote longwave and shortwave, respectively.

## 4.1 Above-canopy radiation budget

Differences in surface albedo can affect air temperature and permafrost thaw (Lawrence and Swenson, 2011; Bonfils et al., 2012). Numerous studies found that shrub tundra has a lower growing season albedo than no-shrub tundra (Chapin et al., 2000a; Thompson et al., 2004; Beringer et al., 2005; Ahrends et al., 2012). Our study agrees with this finding although the albedo difference between dwarf shrubs and wet sedges was small (Table 2). The mean growing season albedo of 0.15 for dwarf shrubs and 0.17 for sedges observed in our study agree well with literature values (Chapin et al., 2000a; Eugster et al., 2000; Ahrends et al., 2012). The measured albedo values are within the range provided by modelling results of the two vegetation types at the same study site (Juszak et al., 2016). While the difference between dwarf shrub albedo and wet sedge albedo is relatively small, the modelling results indicate that the albedo of other vegetation types present at the field site varies more strongly. Juszak et al. (2016) found albedo variation between 0.18 for dry sedge wetland and 0.09 for waterlogged sedges. Therefore, the above canopy radiation budget is more variable if multiple vegetation types are taken into account and may influence the permafrost more strongly. The low values we measured for PAR reflectance (Fig. 7b) are in the same range as values measured by Lloyd et al. (2001) on an Arctic palsa mire.

At our study site, the wet sedge canopy was almost twice as tall as the dwarf shrub canopy (Fig. 3). Furthermore, the plant area index of wet sedges including dead leaves was almost double of the plant area index of dwarf shrubs. In general, taller and denser canopies trap light more efficiently and thus have a lower albedo (Oke, 1987). This is not the case in our comparison between dwarf shrubs and wet sedges, which may be due to two reasons. First, the wet sedge canopy comprises light-coloured standing dead leaves (Chapin et al., 2000a). Second, leaf and wood angle distributions affect canopy reflectance (Verstraete, 1987; Asner, 1998). While dwarf shrubs may have a spherical leaf angle distribution (Juszak et al., 2014), wet sedges likely have erectophil leaves. As compared to the dry summer of 2013, the albedo was lower by 0.01 on dwarf shrubs and by 0.03 on sedges during the wetter summer of 2014. In the wet year, standing water remained at the sedge patch throughout the growing season. Low albedo values on wet sedge locations, especially with standing water, have been reported in literature (Lafleur et al., 1997; Langer et al., 2011; Gamon et al., 2012).

Shrubs are associated with earlier snowmelt and thus decreased spring albedo (Sturm et al., 2005; Pomeroy et al., 2006). In spring 2014, the snow melted ten days earlier on the dwarf shrub patch as compared to the wet sedge patch. As found by Chapin et al. (2005) and Sturm et al. (2005), in our study snowmelt timing was far more important for the overall energy budget than the growing season albedo difference (Fig. 7a). Apart from the large albedo difference in this period, high values of incoming shortwave radiation end of May and beginning of June contributed to the effect. However, at our study site, the earlier snowmelt at the shrub location was not primarily due to branches exposed above the snow surface (Sturm et al., 2005; Pomeroy et al., 2006), but rather due to the thinner snow cover. The snow cover levelled out some of the micro-topography. Thus more snow accumulated in the sedge depression (71 cm) than on the elevated dwarf shrub patch (32 cm, Table 1). This is in contrast to observation from study sites with taller shrubs which trap snow and thus lead to a deeper snow pack (Sturm et al., 2001a; Liston et al., 2002).

We found that clouds reduced the diel albedo by 0.01 (dwarf shrubs) to 0.03 (sedges). These values agree well with the value of 0.02 stated in Eugster et al. (2000) for vegetated and unvegetated tundra. As clear-sky and cloudy-sky albedo differ most at large solar zenith angles, cloud cover reduced the albedo most strongly in the late growing season. The average growing season albedo is likely to decrease in case of increased cloud cover in the future (Chapin et al., 2005; Wang and Key, 2005b; Vavrus et al., 2009). Cloud cover changed over the growing season. Due to the strong dependence of albedo on cloud cover, we could not quantify temporal albedo trends within the growing season that may be caused by soil moisture or vegetation phenology.

While dwarf shrubs and sedges differed in the shortwave radiation budget, the growing season net radiation was similar (Table 2, Fig. 10). On one hand, the dwarf shrub canopy–soil system absorbed more shortwave radiation, on the other hand it emitted more longwave radiation as daily maximum soil temperatures were higher. However, in accordance with Rouse (2000) we found that during the snowmelt period net radiation strongly depended on the snow cover. The different snow melt dates of the vegetation types affected the growing season length locally which may influence the tundra carbon cycle, via respiration, primary production, and methane exchange.

## 4.2 Soil shading

Soil shading by shrubs has been suggested as major factor mitigating permafrost thaw at the local scale (Blok et al., 2010). However, unlike for forests or crops, shading by tundra vegetation has rarely been measured. In the shortwave range, we found an average growing season transmittance of 0.36 below dwarf shrubs and 0.28 below sedges (Table 2). These values are in the same range as modelling results of dwarf shrubs and wet sedges by Juszak et al. (2016). In comparison with other vegetation types present at the site, the transmittance of both types is small. The low transmittance is caused by the high plant area index of dwarf shrubs and wet sedges as compared to other types. In the PAR range, dwarf shrubs transmitted on average 25% and sedges only 3% (Fig. 8c). The dwarf shrub PAR transmittance was in the same range as values by Juszak et al. (2014) measured at the same field site. Williams et al. (2014) measured PAR transmittance below tundra shrubs of 70–100 cm height, two to three times taller than the dwarf shrubs in our study (Fig. 3a). They found a PAR transmittance of about 0.2, which is the lower boundary of the range of values we obtained. The extremely low values of PAR transmittance below sedges were partly due to the lower measurement height of PAR as compared to shortwave transmittance. The shortwave transmittance sensor of 34 mm height was placed above the early summer maximum water level. The PAR sensor of 16 mm height was inserted as low as possible above the current wet litter or water surface. The standing dead leaves are a major component of the wet sedge vegetation (Fig. 1c) and account for most reflection and absorption (Fig. 8c) (Caldwell et al., 1974; Chapin et al., 2000a).

The reference level at the dwarf shrub site is above the shrub litter, which forms a thin, compact layer on the ground with more heavily degraded litter at the bottom and more loose, recent litter towards the top. Unlike the shrub canopy, the sedge canopy includes lots of standing dead material (Table 1). We defined the reference level for canopy transmittance and soil heat flux below this light-coloured layer but above the wet, dark-coloured, and compact litter. This distinction between standing dead leaves and wet litter is useful because of the different structure of both layers. Standing dead leaves form a 12–34 cm thick layer with arching dry leaves and large air spaces. Below this layer, more compacted, older, and usually water-saturated litter forms a continuous surface. In case of shrub or sedge litter on the ground, energy can be transferred from the litter to the soil through heat conduction. Therefore, the thermal properties of the litter can be treated similar to the thermal properties of the soil. For wet sedges, the thermal properties of the dry, standing dead leaves are less important as the leaves are surrounded by air and heat conduction will be dominated by the air. Thus heat conduction through the standing dead leaves may be limited and energy convection or radiation through the standing dead layer may be more important. Therefore we argue that for energy budget considerations shrub litter and wet litter of sedges can be treated analogously to soil, while sedge standing dead leaves resemble more green leaves and have to be treated as part of the above-ground canopy.

Clouds decreased soil shading of both vegetation types, especially at large solar zenith angles (Fig. 8a, b). The Williams et al. (2014) study on Arctic shrubs did not find a dependency of canopy transmittance on diffuse or direct radiation. However, we found that the effect was strongest for large solar zenith angles and Williams et al. (2014) measured at smaller solar zenith angles, around noon, and more than 2° lower latitude. The strong dependency of clear-sky sedge transmittance on sun angle can be attributed to the vertical orientation of the leaves. In addition to leaf angle distribution, plant area index and canopy height can cause differences between the vegetation types. In general, direct radiation transmittance decreases for large solar zenith

angles as the path through the vegetation lengthens. For both vegetation types, transmittance was more variable under clear-sky conditions, which indicates an additional dependency on the solar azimuth angle for specific locations. Spatial variability of transmittance is thus related to spatial inhomogeneity of canopy structure. The higher transmittance of diffuse light as compared to direct light at large solar zenith angles has been measured in a number of studies (e.g. Eck and Deering, 1992; Promis et al.,

2009; Dengel et al., 2015). However, although the canopies shade less efficiently at cloudy conditions than during clear-sky hours, clouds reduce the absolute amount of transmitted shortwave radiation. Thus if the cloud cover increases, less shortwave radiation warms the soil directly.

## 4.3   Soil heat flux and permafrost active layer

Increasing active layer thickness can lead to substantial carbon emissions from permafrost soils (Schuur et al., 2009) and thus

positively feeds back to climate warming (Field et al., 2007). We found a $17\,\mathrm{cm}$ shallower active layer and $6.2\,\mathrm{W\,m^{-2}}$ lower soil heat flux at $10\,\mathrm{cm}$ depth below dwarf shrubs as compared to wet sedges (Fig. 9a, Table 2), which agrees well with other studies (Anisimov et al., 2002; Walker et al., 2003b; Blok et al., 2010). We evaluated possible causes of this difference between vegetation types. The dwarf shrub canopy reflected less shortwave radiation and transmitted more to the moss or soil surface below. Thus, if the shortwave radiation budget was the major driver, a higher soil heat flux could be expected below the dwarf

shrubs. The outgoing longwave radiation was slightly higher at the dwarf shrub patch. Thus the above-canopy net-radiation was almost equal at both vegetation types (Table 2), and hence differences in resulting energy fluxes (sensible, latent, and ground heat flux) were purely internally controlled by the vegetation structure and activity and by soil processes.

Several Arctic studies found similar or less evapotranspiration in low shrub tundra as compared to wetland tundra (Eugster et al., 2000; Rouse, 2000; Eaton et al., 2001; McFadden et al., 2003). At our field site, dwarf shrub LAI was smaller than

sedge LAI (Fig. 3b) and the shrub soil was much drier. Thus, we do not expect more energy loss due to evapotranspiration of dwarf shrubs. Despite the close vicinity of wet sedges and dwarf shrubs, the frequent wind, and the similar average top soil temperatures, air temperature above the wet sedges was about $0.6\,°\mathrm{C}$ colder than above dwarf shrubs (Figure 4e). This finding is consistent with the hypothesis of higher evapotranspiration at the wet sedge site.

The higher ground heat flux at the wet sedges may have lead to reduced sensible heat flux (McFadden et al., 1998). Although

the average top-soil temperatures were very similar below both vegetation types (Table 2) and the gradient between air and soil temperature was smaller at wet sedges (Fig. 4e), the heat flux towards the sedge soil was larger.However, the same gradient between cold soil and warm air temperature may lead to higher flux below sedges as the thermal conductivity of the water-logged sedges soil was about five times higher than of the peaty dwarf shrub top soil (Table 1). The heat capacity below wet sedges was more than six times the value measured below dwarf shrubs (Table 1). This stronger energy sink may have further

enhanced the soil heat flux below sedges. Further, lateral soil water flux from the elevated shrub patches towards the lower lying sedge patches may contribute to the increased soil heat flux in sedge patches. Williams and Quinton (2013) also found that altered moisture conditions were more important for permafrost thaw than the shortwave radiation budget along linear disturbances in a boreal forest.

Mosses and lichen are important components of tundra canopies. Peat mosses contribute significantly to the different soil thermal properties which we observed between dwarf shrubs and wet sedges. Mosses can insulate the soil from air temperature (Beringer et al., 2001; Blok et al., 2011). Furthermore, mosses can alter microtopography and thus modify drainage and moisture conditions (Gornall et al., 2007). As evapotranspiration is affected by mosses in multiple ways (Vourlitis and Oechel, 1999), they need to be considered when modelling energy, water and carbon exchange of Arctic and boreal ecosystems (Launiainen et al., 2015). Another important aspect is soil albedo. Mosses and lichen influence the albedo of the background below the canopy (Stoy et al., 2012). In this way, they do not only affect landscape albedo, but also the amount of radiation absorbed by the soil and the moss layer (Stoy et al., 2012). The wet litter surface below sedges had a low albedo, possibly less than the litter and moss surface below dwarf shrubs. Model results by Juszak et al. (2014) indicate a surface albedo below dwarf shrubs of 0.17 at our site. Given the average growing season transmittance of $64\,\mathrm{W\,m^{-2}}$, the dwarf shrub soil may absorb around $53\,\mathrm{W\,m^{-2}}$. With a low soil albedo, the sedge soil may have absorbed a greater fraction of the transmitted shortwave radiation ($49\,\mathrm{W\,m^{-2}}$ on average) than the dwarf shrub soil, an effect that may partly compensate the more efficient shading. However, the above and below canopy radiation budget of dwarf shrubs and wet sedges is relatively similar which implies that other factors contribute more strongly to the differences in soil heat flux. Other vegetation types present at the site vary more strongly in albedo and reveal much higher transmittance (Juszak et al., 2016). Therefore, the radiation budget may explain more of the spatial variability in active layer thickness if more vegetation types are considered.

In summary, differences in net radiation between dwarf shrubs and wet sedges are smaller than expected, and clearly additional driving forces besides canopy–radiation interactions must be considered for explaining soil heat flux and active layer thickness in future studies, namely soil albedo and soil thermal conductivity (Fig. 10).

## 5 Conclusions

Our field data show that permafrost thaw was lower below tundra dwarf shrubs as compared to wet sedges, but not as a result of increased soil shading. Neither the above-canopy radiation budget nor soil shading explained the spatial differences in active layer thickness. Despite lower shortwave reflectance and higher transmittance by the dwarf shrubs, the soil below dwarf shrubs showed a smaller heat flux and a shallower active layer than below sedges. We found that the differences in snow melt timing were more important for the shortwave radiation budget than growing season albedo differences between the two vegetation types. Cloud cover reduced albedo and soil shading of both vegetation types but more strongly so for sedges. Standing dead leaves accounted for most of the soil shading of the sedge canopy. Soil properties, such as soil albedo and thermal conductivity, appear to be more important than the direct effect of the above-ground vegetation layer. Our results highlight the complexity of the atmosphere–vegetation–permafrost interaction. Future studies will need to incorporate plant traits, such as green, woody, and dead biomass, soil properties, as well as spatial patterns of vegetation types. These variables may be key controls of the potential feedback between vegetation changes and permafrost thaw and deserve more attention to understand the complex interactions between tundra ecosystems and climate.

*Author contributions.* I. Juszak and G. Schaepman-Strub conceived the study. I. Juszak carried out the field measurements and analysed the data with input from G. Schaepman-Strub and W. Eugster. I. Juszak prepared the manuscript with contributions from all co-authors.

The time series data used for this work are available at https://doi.pangaea.de/10.1594/PANGAEA.860561.

*Acknowledgements.* We thank Trofim C. Maximov and his group at the Siberian Branch of the Russian Academy of Science for supporting our field work. We thank Joseph P. McFadden, UC Santa Barbara, and Luca Belelli Marchesini, VU Amsterdam, for fruitful discussions on earlier versions of this manuscript. Furthermore, we thank Samuli Launiainen and an anonymous referee for their helpful comments and suggestions. This work was supported by the Swiss National Science Foundation through project grant 140631 and by the Netherlands Organisation for Scientific Research (NWO Vidi-grant 864.09.014). We were further supported by the University of Zurich through the University research priority programme on Global Change and Biodiversity (URPP GCB).

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
