# Peer review of "Contrasting radiation and soil heat fluxes in Arctic shrub and wet sedge tundra"

_Biogeosciences, 2016_

## Referee Comment (RC1) · Anonymous Referee #1 · 10 Mar 2016

General comment: This study focus on the issue of the heat exchange between the land surface (Tundra ecosystems) and atmosphere over the Altaic region. The authors conducted this study by comparing the surface energy budget at the sedges site and the shrub site. The design of this study is quite straight forward to justify environmental drivers to contrasting the energy into the permafrost soil layer. The authors concluded that the surface background effect (soil albedo) is more important than the shading effect contributed from vegetation covers, while the most of energy was transferred into the soil layer due to the heat conduction (from warm air to cool soil) instead of the radiative absorption by the vegetation cover itself. This is an interesting finding for evaluating the land-atmosphere energy exchange under warm overlying air, high wind speed, weak radiation, and frozen soil conditions. The result would be useful for the future SVAT modelling over the Artic region, which is a recent theme of the

Biogeoscience research in relation to the topic of the climate warming. I suppose that this manuscript could be publish in the journal finally.

However, the information of long-wave radiation did not well presented in this manuscript, I recommend the authors can analysis the long-wave radiation balance\budget during daytime and nighttime. I would also like to understand the diurnal course of temperature gradient between the soil and overlying air parcel to elucidate the direction of sensible heat flux. Besides, I noticed that the authors mentioned the soil moisture condition at sedges site was always under the saturated condition, but the evapotranspiration at the sedges site was suggested to be higher than that at the shrub site. This implies that the soil moisture at sedges site was replenished/affected by lateral water flux, which could also transport heat from other regions such as upland area with the shrub vegetation cover. I recommend that the authors can cite relative studies regarding to the lateral water flux and heat transport at top soil layer over this region or the authors can add an extra analysis of soil moisture by using the soil moisture depletion approach.

Specific comments:

P1 line7: How to define the active layer thickness in this study?

P1 line14 to 15: The authors should provide the evidence such as soil moisture information, soil albedo to support this conclusion. I can't find the approach that the authors conduct the observation of soil albedo measurement throughout the manuscript. Would you please indicate that how to measure the soil albedo? Does it also parameterize as a function of soil moisture change or solar zenith angle?

P3 line26: Please remove "e.g." for the consistence.

P4 line4: Would you please also provide the information of above ground biomass at the sedges and shrub sites? It would be nice to show this information to readers for the comparison.

P4 line9 & line 10, P5 line2-3; P17 line10-11 Please check the unit of the thermal conductivity and heat capacity, is it correct?

P12 Figure 7 and Figure 8: The information contains in the Figure 8 which is largely repeated from the Figure 7, thus I recommend to remove the Figure 8.

P13 line14: "Depended on soil properties", What kind of soil properties, thermal conductivity, porosity, or soil moisture?

P13 Figure 9: Would you please apply the soil moisture depletion approach (Michelakis et al. 1994) to calculate the reference evapotranspiration rate or apply the Priestley - Taylor approach (Priestley and Taylor 1972) to calculate the evapotranspiration rate limited by a correction function based on LAI or soil moisture conditions?

Reference: Ref1: Michelakis, N.I.C., Vouyoucalou, E. and Clapaki, G. 1994. Soil moisture depletion, evapotranspiration and crop coefficients for olive trees cv. kalamon, for different levels of soil water potential and methods of irrigation. Acta Hortic. 356, 162-167 Ref2: Priestley, C.H.B., and R.J. Taylor. 1972. On the assessment of surface heat flux and evaporation using large-scale parameters. Mon. Weather Rev., 100:81-82

P15 line18: I was confused by this sentence, "strong cloud impact on albedo masked other temporal trends within growing season ...." To my knowledge, the calculation of surface albedo (vegetation + soil background) can be separated into two parts (visible + near infrared). The reflectance (albedo) from near infrared is more sensitive to the canopy structure (Otte et al. 2014), and albedo are often parametrized as a function of solar zenith angle in the radiative transfer process. Would you please use this concept to explain your finding in a logic way?

Reference: Otte et al., 2014: Forest summer albedo is sensitive to species and thinning: how should we account for this in Earth system models? Biogeosciences, 11, 2411–2427.

---

## Referee Comment (RC2) · S. Launiainen (Referee) · 16 Mar 2016

General comments:

The study explores vegetation controls on surface radiation budget and soil heat flux in Arctic tundra ecosystem in Siberia. The topic is interesting and timely, since it has direct connection to permafrost dynamics and climate feedbacks of vegetation changes in the Arctic region. The topic fits also well into the scope of Biogeosciences.

This is a solid experimental study which novelty arises from subject of the study, not from the study design. The field experiment and measurements seem to be designed and performed carefully, and data-analysis and discussion is adequate. To gain further understanding, the experimental results need to be considered together / analyzed with soil-vegetation-atmosphere transfer models. Inclusion of relatively simple model

schemes (e.g. canopy radiative transfer, soil heat balance) would allow explaining the empirical findings using theoretical grounds. I understand this may be unrealistic for the current study, and thus encourage the authors to publish the dataset to allow its use by the modeling community. From modeling perspective, it is unfortunate that e.g. soil surface temperature, wind speed, and turbulent heat fluxes from the combined shrub/sedge ecosystem were not measured at the study site?

The manuscript is transparent and well written, and I consider it as useful contribution to our understanding of controls of soil thermal regime in Arctic tundra. I recommend the paper to be published in Biogeosciences after minor revisions.

Specific comments:

P7 L9-11: Please clarify: 'In order to reduce the solar angle influence, we took daily average fluxes of K and L to compute Rn, alpha and T for the analysis of vegetation type and cloud cover effect.' Later, you show how you calculate e.g. cloud cover for each 10min period, and show ln the solar angle dependency of above parameters separately for clear-sky and cloudy conditions (Figs. 5, 7 &8 ) so is there a mistake in text?

As Reviewer 1, I would expect to see e.g. ensemble diurnal cycles of Rn/Kdn, Lnet, alpha and T. Such a figure could replace Fig. 8 which I consider unnecessary since same information is given already in Fig. 7.

P8 L 15 – 25: Please clarify: What are the total LAI's & WAI (woody area index) above sub-canopy radiation sensors for shrub and sedge –sites. These are needed to interpret canopy transmittance (T). Does the LAI of sedge (1.4 +/- 0.3 m2m-2) include the dead standing leaves? If not, what is their LAI?

P10 L 18-19: The differences in transmittance (clear-sky vs. cloudy) can be explained by different plant-area index (LAI+WAI) of shrubs and sedges, and partly by different leaf orientation (spherical vs. vertical leaf angle distribution).

P11 Fig 5: Maybe consider showing Rn relative to incoming global radiation (Kdn)? Now you compare apples and oranges since global radiation varies strongly between cloudy and clear-sky conditions, and with solar zenith angle.

P12 L5: Would be interesting to see ensemble diurnal cycles of soil heat flux, soil-air temperature gradient and radiation (net short- and longwave) for core growing season. See also later comment.

P14 L9 and P15 L30-31: Plant-area index (PAI) is the main control of light extinction within canopies, not canopy height. Of course, taller canopies have often high PAI, and also more complex architecture (at shoot, branch, canopy levels) that enhance absorption of solar radiation compared to shallow vegetation.

This is a section where use of a simple canopy radiation models (e.g. Spitters, 1986; Zhao & Qualls, 2005) would have been beneficial to back up the discussion.

P17 L3 – 18: This is a section where use of soil heat transfer models could be of great help. You arrive into conclusion that net radiation at the soil surface is not a major cause of shrub/sedge plot difference in soil heat flux. So, one could use same upper forcing for both plots (shrub/sedge) and ask how much the different soil properties (thermal conductivity, heat capacity that are measured) explain the observed difference in soil heat flux at 10 cm depth.

Deeper non-saturated layer and 5 cm thick moss cover at shrub site are likely to act as an insulating media. This restricts heat conduction to deeper layers reducing the heat flux measured at 10cm depth. As consequence, the diurnal variability of top soil temperature (measured at 4cm depth?) and top soil heat storage change at shrub site should be much greater than at the sedge site.

If this is not the case, then it is the soil surface energy budget that explains the difference. Since you think net radiation is not significantly different, this would mean that net turbulent energy transfer (sensible + latent heat) should be stronger at the shrub

site. Because shrub canopy is sparser than sedge canopy, I would expect that eddy diffusivity (exchange coefficient) is larger for shrub. Without detailed model it is not easy to speculate what happens to latent heat flux from moss-dominates soil surface (shrub-site) vs. moist peat/litter surface at the sedge sites. The analysis of soil – air temperature difference and its seasonal / diurnal variability should, however, give some indication of possible differences in the sensible heat exchange.

See e.g. Stoy et al. (2012) if interested on possibly significant impacts of moss cover /moss type on soil heat flux and temperature, and Launiainen et al. (2015) for an example of modeling soil-moss-air energy exchange below plant canopies.

Technical corrections:

P2 L15: 'Vegetation alters the radiation budget and turbulent energy fluxes at the soil surface...'

P4 L 7: 'volumetric soil moisture'

P4 L8 – P5 L4: Please note here that you measured the thermal conductivities and heat capacities at the sites.

P4 L5: At first read I was expecting 'energy fluxes' to include also sensible and latent heat. Please be more exact in the 1st sentence of the chapter; you measured only radiation above and below the canopy, and soil heat flux at 10cm depth.

P13 L9: Include definition of active layer depth for a general reader?

P16 L 22: Spatial variability of transmissivity is thus related to spatial inhomogeneity of canopy structure.

P16 L31: drivers of processes but causes of differences

P17 L2: ... and activity, and by soil processes (not or)

References:

Launiainen, S., Katul, G. G., Lauren, A., & Kolari, P. (2015). Coupling boreal forest $CO_2$, $H_2O$ and energy flows by a vertically structured forest canopy–soil model with separate bryophyte layer. Ecological Modelling, 312, 385-405.

Spitters, C. J. T. (1986). Separating the diffuse and direct component of global radiation and its implications for modeling canopy photosynthesis Part II. Calculation of canopy photosynthesis. Agricultural and Forest meteorology, 38(1), 231-242.

Stoy, P. et al. 2012. Temperature, heat flux and reflectance of common subarctic mosses and lichens under field conditions: Might changes to community composition impact climate-relevant surface fluxes? Antarctic, Arctic, and Alpine Res. 44: 500-508.

Zhao, W., & Qualls, R. J. (2005). A multiple layer canopy scattering model to simulate shortwave radiation distribution within a homogeneous plant canopy. Water resources research, 41(8).

———————————————

---

## Author Comment (AC1) · 7 May 2016

We thank the first reviewer for his/her positive statement and his/her interest in our results.

GENERAL COMMENTS

1. However, the information of long-wave radiation did not well presented in this manuscript, I recommend the authors can analysis the long-wave radiation balance/budget during daytime and nighttime.

   We originally focused our analysis on net radiation, albedo, and transmittance. However, we agree that the longwave radiation budget may be interesting for some readers. Outgoing longwave radiation is higher above dwarf shrubs as

compared to above wet sedges due to higher soil temperatures at day time. We will add the corresponding diel cycles to a new figure (see Figure 1 below) replacing the former Figure 8 but appearing earlier in the manuscript.

2. I would also like to understand the diurnal course of temperature gradient between the soil and overlying air parcel to elucidate the direction of sensible heat flux.

We agree with the reviewer and will add the information in the new figure (see Figure 1 below). We found that air temperature at $1.7\,\mathrm{m}$ above the soil surface was similar above both vegetation types, probably due to the small distance between the patches and the wind that we commonly observed. On average, the air temperature above the wet sedges was about $0.6\,°\mathrm{C}$ colder than above dwarf shrubs. This effect reduced the gradient between air and soil temperature for wet sedges. However, at both vegetation types air temperature was warmer than soil temperature, which indicates that the sensible heat flux was on average directed towards the soil during the growing season.

3. Besides, I noticed that the authors mentioned the soil moisture condition at sedges site was always under the saturated condition, but the evapotranspiration at the sedges site was suggested to be higher than that at the shrub site. This implies that the soil moisture at sedges site was replenished/affected by lateral water flux, which could also transport heat from other regions such as upland area with the shrub vegetation cover. I recommend that the authors can cite relative studies regarding to the lateral water flux and heat transport at top soil layer over this region or the authors can add an extra analysis of soil moisture by using the soil moisture depletion approach.

Below dwarf shrubs, soil moisture decreases during the course of the growing season. Below wet sedges, we observed that the water level decreased but remained at or above the soil surface. We agree with the reviewer that this indicates

lateral water influx. Indeed the wet sedge vegetation is located about $0.7\,\mathrm{m}$ below the adjacent shrub patches. Thus lateral water fluxes can contribute to drying the shrub patch and replenish the soil water below wet sedges. In addition to lateral water fluxes, the active layer plays a key part in permafrost environments. Thawing of frozen water within the active layer depth may supply water. This water supply depends on the moisture conditions at freeze-back in the previous year.

We agree that the mentioned lateral water flux can transport energy towards the wet sedges. However, this energy is difficult to quantify and few studies measured lateral water and heat flux in patterned tundra. Boike et al. (2013) estimated that about 10% of the summer precipitation was converted to runoff at a polygonal tundra site in the Lena river delta, northeast Siberia. Helbig et al. (2013) found that lateral water fluxes were important and variable water residence times contributed to keeping a high water level in the surface drainage network. As we do not have further information on the lateral fluxes at our study site, it would be too speculative to provide a quantitative estimate for lateral heat transport. However, we will add clarifying text to indicate this possibility of lateral heat fluxes.

We investigated whether the approach by Michelakis et al. (1994) may provide additional insights at our site. This approach was developed for olive trees which is in stark contrast to vegetation growing on permafrost soils where only a shallow active layer with low temperatures (and frozen ground beneath) is available. In addition to rainfall and runoff - which is not available with the necessary accuracy and resolution for our study site - also estimates of water gains due to increasing thaw depth over the season would be required, another variable that is not available in sufficient quality for that task. Hence, although this is a welcome and relevant suggestion, we cannot offer such an additional analysis.

However, air temperature may provide an independent indication of evapotranspiration. Cooler air temperature suggests, that more energy is used for evapotranspiration as compared to sensible heat flux. Despite the close vicinity of the wet

sedges and the dwarf shrubs, the frequent wind, and the similar top soil temperatures, air temperature above the wet sedges was about $0.6\,°C$ colder than above dwarf shrubs. This finding is consistent with the hypothesis of higher evapotranspiration at the wet sedge site.

SPECIFIC COMMENTS

**P1 line7** How to define the active layer thickness in this study?

The active layer thickness is the thaw depth end of summer measured from the surface of soil, mosses, or flat litter to the top of the frozen ground. We will include a description at the beginning of the methods section.

**P1 line14 to 15** The authors should provide the evidence such as soil moisture information, soil albedo to support this conclusion. I can't find the approach that the authors conduct the observation of soil albedo measurement throughout the manuscript. Would you please indicate that how to measure the soil albedo? Does it also parameterize as a function of soil moisture change or solar zenith angle?

We measured soil moisture below both vegetation types and included this information in the study site description. We agree that soil albedo is a function of soil moisture and possibly solar angle. However, we do not have reliable measurements of soil albedo and are thus not able to include it quantitatively in the manuscript. An indication of soil albedo below dwarf shrubs is provided in Juszak et al. (2014). In that paper, hemispherical-conical reflectance was measured with $1\,nm$ spectral resolution on plots with removed shrub cover at the same field site. Using a radiative transfer model Juszak et al. (2014) estimated a broadband soil albedo of 0.17 for conditions around solar noon (as mentioned in the discussion Section 4.3). This value is not the bare soil albedo but mostly the albedo of shrub

litter and mosses. The background below wet sedges is standing water and saturated, dark litter. We do not have albedo measurements of this water and litter layer. However, wet soil and water at high sun zenith angles (when most energy reaches the ground) generally have low albedo values (Oke, 1987).

**P3 line26** Please remove 'e.g.' for the consistence.

We will replace 'e.g.' with 'mainly'. 'Mainly' is necessary because several species of willows are abundant in the area.

**P4 line4** Would you please also provide the information of above ground biomass at the sedges and shrub sites? It would be nice to show this information to readers for the comparison.

We will add the measured dry biomass values to the study site description.

**P4 line9 & line 10, P5 line2–3; P17 line10–11** Please check the unit of the thermal conductivity and heat capacity, is it correct?

We use the unit $W\,m^{-1}\,K^{-1}$ for thermal conductivity as it is commonly done. We use volumetric heat capacity in $MJ\,m^{-3}\,K^{-1}$. We will add the term 'volumetric' to be more specific. The values of dwarf shrubs and wet sedges agree well with the values for dry peat and still water in Oke (1987), respectively.

**P12 Figure 7 and Figure 8** The information contains in the Figure 8 which is largely repeated from the Figure 7, thus I recommend to remove the Figure 8.

The figure will be removed.

**P13 line14** 'Depended on soil properties', What kind of soil properties, thermal conductivity, porosity, or soil moisture?

The soil heat flux depends on several soil properties. However, we mainly meant soil thermal properties.

**P13 Figure 9** Would you please apply the soil moisture depletion approach (Michelakis et al., 1994) to calculate the reference evapotranspiration rate or apply the Priestley – Taylor approach (Priestley and Taylor, 1972) to calculate the evapotranspiration rate limited by a correction function based on LAI or soil moisture conditions?

As described above in the general comments, the soil moisture depletion approach by Michelakis et al. (1994) is not applicable at our study site due to the lack of data.

The Priestley–Taylor approach (Priestley and Taylor, 1972) requires less input data. Evapotranspiration $E$ is calculated from $E = \alpha \frac{s}{s+\gamma}(R - G)$ where $s$ and $\gamma$ are a function of surface temperature and $\alpha$ depends on water availability, vegetation activity and saturation deficit in the atmosphere. We measured similar net radiation ($R$) and surface temperatures at both vegetation types. Thus the elevated soil heat flux $G$ below wet sedges reduces evapotranspiration as compared to the dwarf shrub patch for a constant $\alpha$. However, $\alpha$ is different between the two vegetation types as the soil below dwarf shrubs is not saturated and the shrubs may close the stomata when they are water limited. Empirical corrections have been developed that relate the parameter $\alpha$ to canopy or soil characteristics (e.g. Flint and Childs, 1991; Sumner and Jacobs, 2005). However, empirical relationships can only be used in conditions which are similar to the calibration conditions. We do not know of any empirical evapotranspiration parametrisation which was calibrated for tundra vegetation. Thus, we prefer to focus the manuscript on the measured data and not include empirical models.

**P15 line18** I was confused by this sentence, 'strong cloud impact on albedo masked other temporal trends within growing season ....' To my knowledge, the calculation of surface albedo (vegetation + soil background) can be separated into two parts (visible + near infrared). The reflectance (albedo) from near infrared is more sensitive to the canopy structure (Otto et al., 2014), and albedo are often parametrized as a function of solar zenith angle in the radiative transfer process.

**Would you please use this concept to explain your finding in a logic way?**

We will change the confusing sentence. We did not calculate the albedo from visible and near-infrared radiation, but we measured broadband shortwave incoming and reflected radiation and calculated albedo from the ratio between the two. We agree, that the effect of clouds on albedo is based on the angles of incident shortwave radiation. While most radiation comes directly from the direction of the sun under clear-sky conditions, clouds diffuse the radiation which consequently comes from all directions. Diffuse radiation commonly lowers the albedo for sun zenith angles above $55\,^\circ$ (Oke, 1987; Yang et al., 2008).

**References**

Boike, J., Kattenstroth, B., Abramova, K., Bornemann, N., Chetverova, A., Fedorova, I., Fröb, K., Grigoriev, M., Grüber, M., Kutzbach, L., Langer, M., Minke, M., Muster, S., Piel, K., Pfeiffer, E.-M., Stoof, G., Westermann, S., Wischnewski, K., Wille, C., and Hubberten, H.-W.: Baseline characteristics of climate, permafrost and land cover from a new permafrost observatory in the Lena River Delta, Siberia (1998 – 2011), Biogeosciences, 10, 2105–2128, doi:10.5194/bg-10-2105-2013, 2013.

Flint, A. L. and Childs, S. W.: Use of the Priestley-Taylor evaporation equation for soil water limited conditions in a small forest clearcut, Agricultural and Forest Meteorology, 56, 247–260, doi:10.1016/0168-1923(91)90094-7, 1991.

Helbig, M., Boike, J., Langer, M., Schreiber, P., Runkle, B. R. K., and Kutzbach, L.: Spatial and seasonal variability of polygonal tundra water balance: Lena River Delta, northern Siberia (Russia), Hydrogeology Journal, 21, 133–147, doi:10.1007/s10040-012-0933-4, 2013.

Juszak, I., Erb, A. M., Maximov, T. C., and Schaepman-Strub, G.: Arctic shrub effects on NDVI, summer albedo and soil shading, Remote Sensing of Environment, 153, 79–89, doi:10.1016/j.rse.2014.07.021, 2014.

Michelakis, N. I. C., Vouyoucalou, E., and Clapaki, G.: Soil moisture depletion, evapotranspiration and crop coefficients for olive trees cv. kalamon, for different levels of soil water potential

and methods of irrigation, Acta Horticulturae, 356, 162–167, doi:10.17660/ActaHortic.1994.356.34, 1994.

Oke, T. R.: Boundary layer climates, Methuen & Co, 2 edn., 1987.

Otto, J., Berveiller, D., Bréon, F.-M., Delpierre, N., Geppert, G., Granier, A., Jans, W., Knohl, A., Kuusk, A., Longdoz, B., Moors, E., Mund, M., Pinty, B., Schelhaas, M.-J., and Luyssaert, S.: Forest summer albedo is sensitive to species and thinning: how should we account for this in Earth system models?, Biogeosciences, 11, 2411–2427, doi:10.5194/bg-11-2411-2014, 2014.

Priestley, C. H. B. and Taylor, R. J.: On the Assessment of Surface Heat Flux and Evaporation Using Large-Scale Parameters, Monthly Weather Review, 100, 81–92, doi:10.1175/1520-0493(1972)100<0081:OTAOSH>2.3.CO;2, 1972.

Sumner, D. M. and Jacobs, J. M.: Utility of Penman-Monteith, Priestley-Taylor, reference evapotranspiration, and pan evaporation methods to estimate pasture evapotranspiration, Journal of Hydrology, 308, 81–104, doi:10.1016/j.jhydrol.2004.10.023, 2005.

Yang, F., Mitchell, K., Hou, Y.-T., Dai, Y., Zeng, X., Wang, Z., and Liang, X.-Z.: Dependence of Land Surface Albedo on Solar Zenith Angle: Observations and Model Parameterization, Journal of Applied Meteorology and Climatology, 47, 2963–2982, doi:10.1175/2008JAMC1843.1, 2008.

[Figure]

**Figure 1** (a) Average diel cycle of above canopy shortwave ($K$) and longwave ($L$) radiation fluxes, (b) above canopy net shortwave ($K_n$), net longwave ($L_n$) and net ($R_n$) radiation, (c) albedo ($\alpha$) and transmittance ($T$), (d) soil temperature ($T_s$) at $4\,$cm depth and soil heat flux ($HF$) at $10\,$cm depth, and (e) air temperature at $1.7\,$m above the soil surface ($T_a$) and gradient between air and soil temperature ($\Delta T$) of dwarf shrubs and wet sedges during the growing season; solar noon at 14:00 local time.

[Figure]

[Figure]

**Fig. 1.** For caption please see above

---

## Author Comment (AC2) · 7 May 2016

We thank Mr Launiainen for the positive evaluation of our manuscript and for his comments and suggestions. We are also grateful for his advice on how our work can be relevant to the modelling community.

GENERAL COMMENTS

1. To gain further understanding, the experimental results need to be considered together / analyzed with soil-vegetation-atmosphere transfer models. Inclusion of relatively simple model schemes (e.g. canopy radiative transfer, soil heat balance) would allow explaining the empirical findings using theoretical grounds. I understand this may be unrealistic for the current study, and thus encourage the

authors to publish the dataset to allow its use by the modeling community.

Using a model would be interesting, but would extend well beyond the scope of our paper. We will hence provided all available meteorological driving variables (radiation fluxes and air temperature) in combination with our canopy and soil data via Pangea to allow for such modelling in follow-up studies.

2. From modeling perspective, it is unfortunate that e.g. soil surface temperature, wind speed, and turbulent heat fluxes from the combined shrub/sedge ecosystem were not measured at the study site?

Ecosystem-scale measurements were done nearby our study location at a flux tower operated by the VU Amsterdam (van der Molen et al., 2007; Parmentier et al., 2011; Budishchev et al., 2014). However, the measurements do not include soil surface temperature. The data are accessible via the European flux database (Russia, Chokurdakh, http://www.europe-fluxdata.eu). Unfortunately, the small size of the vegetation patches and the permafrost conditions did not allow vegetation type specific measurements of sensible and latent heat flux. Such measurements would have been useful for understanding the energy budget of the different vegetation types.

SPECIFIC COMMENTS

**P7 L9–11** Please clarify: In order to reduce the solar angle influence, we took daily average fluxes of K and L to compute Rn, alpha and T for the analysis of vegetation type and cloud cover effect. Later, you show how you calculate e.g. cloud cover for each 10min period, and show In the solar angle dependency of above parameters separately for clear-sky and cloudy conditions (Figs. 5, 7 & 8 ) so is there a mistake in text?

The solar angle influence was reduced to estimate the mean vegetation and cloud effects as shown in Table 1 (and in the text). On the other hand, we analysed the

effect of solar angles on the radiation budget as shown in the Figures 5, 7 & 8. We will change the paragraph to better distinguish between the two methods.

As Reviewer 1, I would expect to see e.g. ensemble diurnal cycles of Rn/Kdn, Lnet, alpha and T. Such a figure could replace Fig. 8 which I consider unnecessary since same information is given already in Fig. 7.

We will remove the former Figure 8 and add a new figure (see Figure 1 below) with the diel cycles.

**P8 L 15–25** Please clarify: What are the total LAI's & WAI (woody area index) above sub-canopy radiation sensors for shrub and sedge-sites. These are needed to interpret canopy transmittance (T). Does the LAI of sedge (1.4 +/- 0.3 m2m-2) include the dead standing leaves? If not, what is their LAI?

For dwarf shrubs, LAI and WAI would ideally have been completely above the sub-canopy sensor. There is a little difference between the two sensor types used for below canopy radiation. The sensor for time series measurements of shortwave transmittance was 34 mm high, roughly 13% of the canopy height of dwarf shrubs. Thus we assume that about 80% of the LAI and WAI are above this sensor. The PAR sensor (Figure 8c, former Figure 7c) is only 16 mm thick and thus should measure the transmittance of almost the complete canopy. The sedge LAI as measured in the field includes only the green leaves. We included this clarification in multiple places in the text now. We also estimated the standing dead sedge leaf area index and found that it was about 1.1 times green leaf area. We found a mistake in the original computation and will update this value to the correct number. However, we cannot include the dead leaves in the area estimate in Figure 3 as it was not measured on the same eight plots as the green leaf area index. Instead, the ratio between green and dead leaves was estimated on three additional plots destructively.

**P10 L 18-19** The differences in transmittance (clear-sky vs. cloudy) can be explained

by different plant-area index (LAI+WAI) of shrubs and sedges, and partly by different leaf orientation (spherical vs. vertical leaf angle distribution).

We agree and will add this to the discussion section.

**P11 Fig 5** Maybe consider showing Rn relative to incoming global radiation (Kdn)? Now you compare apples and oranges since global radiation varies strongly between cloudy and clear-sky conditions, and with solar zenith angle.

Figure 5 was not meant for a comparison between albedo and net radiation but rather to show both quantities independently. Figure 2 below shows the net radiation normalised with incoming shortwave radiation in comparison with the original figure panels. Normalised net radiation decreases with higher solar zenith angles in the same way for both vegetation types.

We would prefer to keep the original Figure 5 instead of replacing the panels a and b with the panels c and d of Figure 2 below as net radiation is commonly used and easier to interpret than normalised net radiation.

**P12 L5** Would be interesting to see ensemble diurnal cycles of soil heat flux, soil-air temperature gradient and radiation (net short- and longwave) for core growing season. See also later comment.

We will add the requested figure (see Figure 1 below).

**P14 L9 and P15 L30–31** Plant-area index (PAI) is the main control of light extinction within canopies, not canopy height. Of course, taller canopies have often high PAI, and also more complex architecture (at shoot, branch, canopy levels) that enhance absorption of solar radiation compared to shallow vegetation. This is a section where use of a simple canopy radiation models (e.g. Spitters, 1986; Zhao and Qualls, 2005) would have been beneficial to back up the discussion.

We agree that canopy absorptance and transmittance strongly depend on PAI while canopy architecture (height and angle distribution) is a secondary control.

We will clarify the sentence in question. Furthermore, we agree that models of canopy radiation transfer can be a valuable tool for understanding the differences between the radiation budget of the two vegetation types. However, we covered radiative transfer modelling of different tundra vegetation types in a second manuscript which is currently under review in *Remote Sensing of Environment*. Hence, it should not be repeated here. Since that paper is not yet formally accepted we did not cite it, but as soon as it becomes citable we will do so. Our modelling results agree well with the field observations. While wet sedge albedo is 0.03 higher than dwarf shrub albedo, sedges transmit 7% less shortwave radiation.

**P17 L3–18** This is a section where use of soil heat transfer models could be of great help. You arrive into conclusion that net radiation at the soil surface is not a major cause of shrub/sedge plot difference in soil heat flux. So, one could use same upper forcing for both plots (shrub/sedge) and ask how much the different soil properties (thermal conductivity, heat capacity that are measured) explain the observed difference in soil heat flux at 10 cm depth. Deeper non-saturated layer and 5 cm thick moss cover at shrub site are likely to act as an insulating media. This restricts heat conduction to deeper layers reducing the heat flux measured at 10cm depth. As consequence, the diurnal variability of top soil temperature (measured at 4cm depth?) and top soil heat storage change at shrub site should be much greater than at the sedge site. If this is not the case, then it is the soil surface energy budget that explains the difference. Since you think net radiation is not significantly different, this would mean that net turbulent energy transfer (sensible + latent heat) should be stronger at the shrub site. Because shrub canopy is sparser than sedge canopy, I would expect that eddy diffusivity (exchange coefficient) is larger for shrub. Without detailed model it is not easy to speculate what happens to latent heat flux from moss-dominates soil surface (shrub-site) vs. moist peat/litter surface at the sedge sites. The analysis of soil –

air temperature difference and its seasonal / diurnal variability should, however, give some indication of possible differences in the sensible heat exchange. See e.g. Stoy et al. (2012) if interested on possibly significant impacts of moss cover /moss type on soil heat flux and temperature, and Launiainen et al. (2015) for an example of modeling soil-moss-air energy exchange below plant canopies.

We agree that modelling of the active layer temperature and heat flux may help understanding the interactions between vegetation, radiation, and soil heat flux. As mentioned above, we will make all necessary data available on Pangaea for in-depth modelling studies. In the text, we will add the moss component and the very helpful considerations made by this reviewer to the discussion, including the named citations.

TECHNICAL CORRECTIONS

**P2 L15** Vegetation alters the radiation budget and turbulent energy fluxes at the soil surface... and turbulent energy fluxes will be added

**P4 L7** volumetric soil moisture volumetric will be added

**P4 L8 – P5 L4** Please note here that you measured the thermal conductivities and heat capacities at the sites. will be added

**P5 L5** At first read I was expecting energy fluxes to include also sensible and latent heat. Please be more exact in the 1st sentence of the chapter; you measured only radiation above and below the canopy, and soil heat flux at 10cm depth. Will be changed

**P13 L9** Include definition of active layer depth for a general reader? General definition will be included in Section 2.1

**P16 L22** Spatial variability of transmissivity is thus related to spatial inhomogeneity of canopy structure. Will be added

**P16 L31** drivers of processes but causes of differences Will be changed

**P17 L2** ... and activity, and by soil processes (not or) Will be changed

**References**

Budishchev, A., Mi, Y., van Huissteden, J., Belelli-Marchesini, L., Schaepman-Strub, G., Parmentier, F. J. W., Fratini, G., Gallagher, A., Maximov, T. C., and Dolman, A. J.: Evaluation of a plot-scale methane emission model using eddy covariance observations and footprint modelling, Biogeosciences, 11, 4651–4664, doi:10.5194/bg-11-4651-2014, 2014.

Launiainen, S., Katul, G. G., Lauren, A., and Kolari, P.: Coupling boreal forest $CO_2$, $H_2O$ and energy flows by a vertically structured forest canopy – Soil model with separate bryophyte layer, Ecological Modelling, 312, 385–405, doi:10.1016/j.ecolmodel.2015.06.007, 2015.

Parmentier, F. J. W., van Huissteden, J., van der Molen, M. K., Schaepman-Strub, G., Karsanaev, S. A., Maximov, T. C., and Dolman, A. J.: Spatial and temporal dynamics in eddy covariance observations of methane fluxes at a tundra site in northeastern Siberia, Journal of Geophysical Research: Biogeosciences, 116, G03016, doi:10.1029/2010JG001637, 2011.

Spitters, C. J. T.: Separating the diffuse and direct component of global radiation and its implications for modeling canopy photosynthesis Part II. Calculation of canopy photosynthesis, Agricultural and Forest Meteorology, 38, 231–242, doi:10.1016/0168-1923(86)90061-4, 1986.

Stoy, P. C., Street, L. E., Johnson, A. V., Prieto-Blanco, A., and Ewing, S. A.: Temperature, Heat Flux, and Reflectance of Common Subarctic Mosses and Lichens under Field Conditions: Might Changes to Community Composition Impact Climate-Relevant Surface Fluxes?, Arctic, Antarctic, and Alpine Research, 44, 500–508, doi:10.1657/1938-4246-44.4.500, 2012.

van der Molen, M. K., van Huissteden, J., Parmentier, F. J. W., Petrescu, A. M. R., Dolman, A. J., Maximov, T. C., Kononov, A. V., Karsanaev, S. V., and Suzdalov, D. A.: The growing season greenhouse gas balance of a continental tundra site in the Indigirka lowlands, NE Siberia, Biogeosciences, 4, 985–1003, doi:10.5194/bg-4-985-2007, 2007.

Zhao, W. and Qualls, R. J.: A multiple-layer canopy scattering model to simulate shortwave radiation distribution within a homogeneous plant canopy, Water Resources Research, 41, doi:10.1029/2005WR004016, w08409, 2005.

[Figure]

**Figure 1** (a) Average diel cycle of above canopy shortwave ($K$) and longwave ($L$) radiation fluxes, (b) above canopy net shortwave ($K_n$), net longwave ($L_n$) and net ($R_n$) radiation, (c) albedo ($\alpha$) and transmittance ($T$), (d) soil temperature ($T_s$) at $4\,\mathrm{cm}$ depth and soil heat flux ($HF$) at $10\,\mathrm{cm}$ depth, and (e) air temperature at $1.7\,\mathrm{m}$ above the soil surface ($T_a$) and gradient between air and soil temperature ($\Delta T$) of dwarf shrubs and wet sedges during the growing season; solar noon at 14:00 local time.

**Figure 2** Dependence of (a,b) net radiation and (c,d) net radiation divided by incoming shortwave radiation on solar zenith angle and cloud cover for (a, c) dwarf shrub and (b, d) wet sedge, growing season mean $\pm$ standard deviation values calculated for $2\,^\circ$ intervals; the dashed lines represent the mean diel value under each condition.

[Figure]

**Fig. 1.** For caption please see above

[Figure]

Shrubs

Sedges

Net radiation (W m$^{-2}$)

Normalised net radiation

Solar zenith angle (°)

Solar zenith angle (°)

**Fig. 2.** For caption please see above

---

## Author Response (AR1)

**Response to Dr Sebastiaan Luyssaert (Associate Editor)**

We thank Dr Luyssaert for the positive evaluation of our manuscript and for his comments and suggestions.

**GENERAL COMMENT**

The manuscript has now been reviewed by two external referees and the handling associate editor. All agreed that the manuscript is acceptable for publication but suggested several points that could be improved. The authors addressed these concerns during the discussion and are encouraged to prepare a revised manuscript by taking the discussion into account.

We thank the reviewers and Dr Luyssaert again for their helpful comments and effort. We addressed all points raised in the discussion and applied all corrections and suggestions that we indicated in our response. The most important improvements are

– rephrasing the objectives as hypotheses;

– modification of Figure 1 to show the different vegetation types more clearly;

– the new Table 1 with details on the different vegetation types;

– shortening of the methods section;

– correcting the information on standing dead leaf area after finding a mistake;

– the new Figure 4 with diel cycles of the most important variables;

– removing the former Figure 8;

– discussing our results in relation with the modelling results that we obtained in a second study;

– discussing the influence of mosses and lateral water flux on the energy budget.

Additionally we clarified details in the text and improved the wording. Please consider our detailed response to all reviewer comments made in the discussion for a more detailed list of changes and explanations.

The time series data is now published on the Pangaea platform https://doi.pangaea.de/10.1594/PANGAEA.860561 (https://doi.org/10.1594/PANGAEA.860561 once the DOI registration is finished), as cited in the revised manuscript. The data from distributed plot measurements as used for Figures 3, 7b, 8c, 9b and parts of Table 1 have been submitted to the Pangaea platform, and will be published after the processing is finished. We will add this second link to the manuscript as soon as it is available (typically within one week).

**ADDITIONAL COMMENTS**

1. Rephrase the objectives as a hypothesis. The current objectives are basically a (very short) summary of the methods but it is not clear what can be learned from this study. Rephrasing the objectives as a hypothesis will overcome repetition and should indicate to the reader what (s)he can learn from the study.

We rephrased the objectives as two hypotheses: "The first hypothesis is that the canopy radiation budget impacts permafrost thaw. The second hypothesis is that weather conditions significantly affect radiation fluxes in Arctic tundra."

2. Fig 1. subplot b is meaningless without a legend for the grey scales in the background. Link the symbols to the vegetation types in subplots c to f, for example, red triangles denote dwarf shrubs as shown in in Fig 1 (e). Looks like that is what you were trying to do but I could not figure it out.

Figure 1b shows an RGB orthomosaic. As we did not classify the vegetation based on the mosaic, we cannot add a legend for the image. However, we added a legend explaining the symbols and re-used the symbol colour for the canopy pictures. Furthermore, we clarified the content of Figure 1b in the caption and added, that shrubs appear darker on the image than sedges.

3. The description of the vegetation is very detailed but the information is not used in support of the findings. Focus on the essential elements/characteristics of the vegetation structure. A small table could also help to present the differences between the sites in a more concise way.

We shortened the study site description and included the most important characteristics in the new Table 1. Most information from the table is used in the discussion to support our findings (such as differences in ALT, canopy height, background properties that affect soil albedo, soil thermal properties). We prefer not to shorten the table content because the anonymous referee requested information on biomass and S. Launiainen (Referee) indicated that the data may be interesting for modelling soil–moss–air energy exchange. For such a modelling approach, information which is less relevant for our study (such as the moss layer thickness) may be beneficial.

4. The decision to report the results in two separate studies, i.e., a "data" and a "model" manuscript weakens the data-manuscripts because some of the hypothesis could have been confirmed or rejected by making use of the model (see also reviewer 2). Where relevant the authors should cite their own modeling study and discuss the consequences.

We agree that modelling requires high-quality data for validation and improvement. Here we, however, focus in the very first step, which is understanding the processes that are able to explain the mechanisms that might be responsible for the empirical findings deduced from field measurements. We think that in this way we can avoid the bias that is introduced by the selection of a specific model which may or may not be the best to represent the relevant processes. By separating our data-based process study from model application we can freely present our findings in a way that would allow all modellers to improve their model. We also use the data for model input and validation in a second study (Juszak et al. (2016), under review), which we now refer to in our discussion, but by separating the process study from the model study we can take a more neutral position with respect to the question which processes a model needs to take into account to represent the conditions that we found.

Please find below the track-change version of the manuscript.

[revised manuscript text omitted]